# Environment Inference for Learning Generalizable Dynamical System

**Shixuan Liu**[1]    **Yue He**[2*]    **Haotian Wang**[1]    **Wenjing Yang**[1]    **Yunfei Wang**[3]
**Peng Cui**[4*]    **Zhong Liu**[5]

[1]College of Computer Science and Technology, National University of Defense Technology
[2]School of Information, Renmin University of China
[3]College of Systems Engineering, National University of Defense Technology
[4]Department of Computer Science and Technology, Tsinghua University
[5]Laboratory for Big Data and Decision, National University of Defense Technology
`szftandy@hotmail.com, hy865865@gmail.com`
`{wanghaotian13,wenjing.yang,wangyunfei,liuzhong}@nudt.edu.cn`
`cuip@tsinghua.edu.cn`

## Abstract

Data-driven methods offer efficient and robust solutions for analyzing complex dynamical systems but rely on the assumption of I.I.D. data, driving the development of generalization techniques for handling environmental differences. These techniques, however, are limited by their dependence on environment labels, which are often unavailable during training due to data acquisition challenges, privacy concerns, and environmental variability, particularly in large public datasets and privacy-sensitive domains. In response, we propose DynaInfer, a novel method that infers environment specifications by analyzing prediction errors from fixed neural networks within each training round, enabling environment assignments directly from data. We prove our algorithm effectively solves the alternating optimization problem in unlabeled scenarios and validate it through extensive experiments across diverse dynamical systems. Results show that DynaInfer outperforms existing environment assignment techniques, converges rapidly to true labels, and even achieves superior performance when environment labels are available.

## 1   Introduction

Data-driven approaches, especially neural networks, offer a powerful alternative or complement to traditional physics-based methods for understanding complex dynamical systems [4]. Neural network-based emulators are particularly valuable for their ability to provide fast, cost-effective approximations of complex simulations [9, 22], making them especially useful in scenarios where the underlying physics are poorly understood or misinterpreted, or where external disturbances are difficult to model [43, 34]. These emulators are adept at handling large sets of variables and solving problems that are challenging for conventional solvers. Recent advancements in deep learning, along with innovative methods for modeling temporal and spatio-temporal systems, have led to a significant increase in applications across various fields, ranging from simple Hamiltonian dynamics to more complex areas like fluid dynamics and climatology [32, 7].

While recent advancements have shown promising results, they often rely on the assumption that abundant, static data are available to satisfy the independent and identically distributed (IID) hypothesis. However, this assumption is frequently violated in practice due to challenges in data collection, associated costs, and environmental changes driven by exogenous factors [24, 26]. Recent work

---

*Corresponding authors

39th Conference on Neural Information Processing Systems (NeurIPS 2025).

in dynamical systems addresses this by introducing a multi-environment setting, where trajectories follow distinct dynamics across environments. These studies developed generalization methods that learn a shared global component while accounting for environment-specific variations, avoiding the limitations of underperforming averaged models [43, 17].

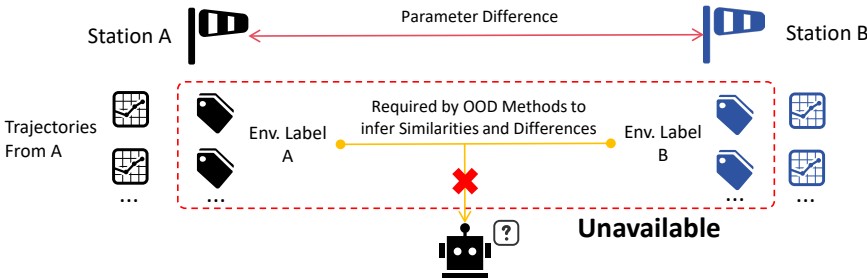

Figure 1: Environment labels, required by current generalization methods, are often unavailable.

Nevertheless, a key limitation of many generalization techniques is their reliance on partitioning datasets across distinct domains or environments, which are assumed to capture underlying variations. These environment labels enable algorithms to identify and exploit both similarities and differences across environments. However, obtaining such environment labels during training is often challenging due to data acquisition difficulties or privacy constraints. For example, in scientific research, data may be collected over time under uncontrolled or unknown conditions [42]. In ecological studies, critical environmental parameters such as temperature or rainfall may vary unpredictably or remain unrecorded. [2]. Similarly, when aggregating data from multiple sources, environment labels are frequently lost or omitted, a common issue in large public datasets [35]. Furthermore, in privacy-sensitive domains like healthcare, finance, or social networking, access to environment-specific information is often restricted [18]. These limitations highlight the need for generalization methods that do not depend on explicit environment labels.

To address the challenge of unknown environment labels, we propose a novel approach that infers environment specifications by leveraging the key insight that trajectories within the same environment share consistent dynamics and exhibit similar prediction losses under the same neural network. This inherent consistency enables us to automatically derive meaningful environment assignments directly from the training data. We introduce an environment inference objective designed for dynamical systems, which minimizes environment-specific prediction losses. Using fixed neural networks, we first infer environments and then iteratively refine these networks with the inferred environments, ultimately learning a generalizable dynamical system.

Our model identifies environment labels directly from mixed trajectories of dynamical systems, facilitating the training of off-the-shelf generalization algorithms in scenarios where such labels are absent. Importantly, our findings demonstrate that inferring environments from mixed sequence data can improve the performance of generalization strategies, even compared to cases where environments are manually assigned.

Our main contributions are as follows:

- We present the first investigation into the challenge of unlabeled environment conditions in the context of learning generalizable dynamical systems, and propose a general framework named DynaInfer that utilizes the prediction loss to accurately infer latent environment labels from mixed sequence data[2].

- We theoretically establish that our algorithm effectively solves the alternating optimization problem without requiring environment labels, demonstrating its capacity to discern heterogeneous environments and infer generalizable mechanisms.

- We examine the efficacy of DynaInfer through experiments in both in-domain settings and adaptation scenarios using three representative dynamical systems. Results confirm that the environment labels assigned by DynaInfer converge rapidly to the true labels.

---

[2]Code is available at `https://github.com/shixuanliu-andy/DynaInfer`

The remainder of this paper is structured as follows. Section 2 clarifies the problem definition. Section 3 introduces our framework and provides the theoretical underpinnings. Section 4 details the experimental setup and discusses the results. Related work is reviewed in Section 5, and Section 6 concludes the paper.

## 2 Problem Definition

### 2.1 Dynamical Systems

We examine dynamical systems determined by unidentified differential equations evolving over time, expressed as,

$$\frac{dx_t}{dt} = f(x_t) \tag{1}$$

where $t \in \mathbb{R}$ is the time index within a time interval $I = [0, T]$, and $x_t$ is a time-variant state within a bounded set $\mathcal{A}$. The evolution function $f : \mathcal{A} \rightarrow T\mathcal{A}$ maps $x_t$ to its temporal derivative in the tangent space $T\mathcal{A}$ and belongs to a class of vector fields $\mathcal{F}$.

In this paper, we consider both ordinary differential equation (ODE) and partial differential equation (PDE). For ODEs, $\mathcal{A} \subset \mathbb{R}^d$; for PDEs, $\mathcal{A}$ represents a $d'$-dimensional vector field within a bounded spatial domain (such as 2D or 3D Euclidean space) denoted as $S \subset \mathbb{R}^{d'}$. The function $f$ characterizes the data distribution of trajectories $\mathcal{T}$. Trajectories initiated from $x_0 \sim p(X_0)$ are computed by integrating the derivatives: $x_t = x_0 + \int_0^t f(x_u)du, \forall t \in I$.

### 2.2 Multi-Environment Dynamical Systems Learning

In contrast to the standard expected risk minimization (ERM) framework, which assumes i.i.d. trajectories, the multi-environment learning problem involves learning trajectories from $M$ different environments. In each environment $e \in [M] = \{1, 2, \ldots, M\}$, the trajectories are governed by unique differential equations described by function $f_e$. Specifically, consider $N$ trajectories $\{x^1, x^2, \ldots, x^N\}$, where each trajectory $x^i$ is associated with an environment $e_i \in [M]$. The dynamics of each trajectory $x^i$ are thus modeled by the differential equation $dx_t^i/dt = f_{e_i}(x_t^i)$. The set of environments for all trajectories is denoted by $\boldsymbol{e} = \{e_1, e_2, \ldots, e_N\} \in [M]^N$.

In multi-environment learning, the goal is to enhance traditional ERM methods by exploiting both the commonalities and disparities across diverse environments. To this end, the dynamics is decomposed into two components: a global component shared across all environments, parameterized by $\theta$, and an environment-specific component, parameterized by $\phi_e$ for each environment $e$. The set of environment-specific parameters is denoted by $\boldsymbol{\phi} = \{\phi_e\}_{e \in [M]}$. Consequently, the dynamics of each trajectory $x^i$ are parameterized by both the universal and environment-specific parameters,

$$\frac{dx_t^i}{dt} = h\left(x_t^i; \theta, \phi_e\right).$$

This parametrization entails a decomposition that can be implemented either functionally or parametrically. The functional decomposition, expressed as $h\left(x_t^i; \theta, \phi_e\right) = f_\theta(x_t^i) + g_{\phi_e}(x_t^i)$, distinguishes between a shared function $f_\theta$ and an environment-specific function $g_{\phi_e}$ [43]. Alternatively, the parametric decomposition integrates the environment-specific parameters directly, formulated as $h\left(x_t^i; \theta, \phi_e\right) = f_{\theta+\phi_e}(x_t^i)$ [17]. Intuitively, the key ingredient for multi-environment learning is that $\theta$ should encapsulate the maximal shared dynamics, whereas $\phi_e$ should exclusively reflect the unique characteristics of each environment $e$ not described by $\theta$. However, directly optimizing both parameters poses an ill-posed problem, often resulting in trivial solutions where the global component learns nothing meaningful. To counteract this, the regularization term $\Omega(\phi_e)$ is introduced to effectively penalize $\phi_e$, thereby facilitating learning in the global component. Consequently, with the information about the environments $\boldsymbol{e} = \{e_1, e_2, \ldots, e_N\}$, the loss function is given by,

$$R_{\boldsymbol{e}}(\theta, \boldsymbol{\phi}) = \sum_{i=1}^{N} \int_{t \in I} \left\| \frac{dx_t^i}{dt} - h\left(x_t^i; \theta, \phi_{e_i}\right) \right\|_2^2 dt + \lambda \sum_{e=1}^{M} \Omega(\phi_e). \tag{2}$$

The first term evaluates the regression precision of the parameterized function $h(\cdot; \theta, \phi_e)$. The ground truth vector field (VF) is not explicitly known and derived from trajectory data. Using the learned VF, a simulated trajectory is generated and used to calculate the regression loss by referring to real trajectories during training. The term $\Omega(\phi_e)$ serves as a regularization term for $\phi_e$, with $\lambda$ controlling the intensity of the regularization.

## 2.3 Environment Inference for Multi-Environment Learning

In many real-world scenarios, the environment label for a trajectory sample is unknown. We aim to infer an environment assignment for each sample that maximizes the model's generalization ability across different environments. To achieve this goal, we reformulate the learning objective into an optimization problem contingent on a specific environment assignment $e$. Specifically, our aim is to learn the environment assignment $\hat{e} = \{\hat{e}^1, \hat{e}^2, \ldots, \hat{e}^N\} \in [M]^N$ for each trajectory to effectively optimize Equation (2). The overall objective is defined as follows:

$$\hat{e}^*, \theta^*, \phi^* = \underset{\hat{e}, \theta, \phi}{\arg\min}\, R_{\hat{e}}(\theta, \phi). \tag{3}$$

In this paper, we explore a particularly challenging scenario where the total number of training environments $M$ is also unknown. We investigate the development of a practical model that maintains favourable performance even when the exact number of true environments is unknown.

# 3 The DynaInfer Framework

In this section, we introduce our framework that operates on field functions without prior domain knowledge, proving especially effective in dynamical systems where exogenous factors are unobserved and in situations where relevant environmental information is unclear or absent. While some clustering methods infer labels for CV data, they operate on finite-dimensional vectors in Euclidean space, which drastically differs from field functions, making them inapplicable.

The optimization challenge in Equation (3) is primarily due to the inherently discrete nature of the environment assignments $\hat{e}$, which take values in the set $[M]$. This discrete categorization impedes the direct application of traditional gradient descent methods, which are typically designed for continuous parameter spaces. To effectively address this challenge, we develop a dual iterative strategy that concurrently updates the environment assignments $\hat{e}$ and the model parameters $\theta, \phi$. The first step in our approach centers on inferring environment labels by analyzing the prediction errors of the trajectories output by the neural network during the current training round. This analysis serves as a diagnostic tool to uncover critical discrepancies that signify distinct dynamical environments. Following this, the second step entails refining the neural network parameters based on the newly inferred environment assignments in an unbiased manner, enabling the neural network to precisely adapt to the unique characteristics of each identified environment. Through this adaptive refinement, our model progressively enhances its accuracy and generalization capability across different dynamic settings. The complete method is detailed in Algorithm 1 and is visually depicted in Figure 2.

---

**Algorithm 1** DynaInfer framework

1: **Input:** Randomly initialized $\theta, \phi = \{\phi_e\}_{e \in [M]}$, assumed number of environments $M$, total rounds $T_r$
2: $\theta^{(0)}, \phi^{(0)} \leftarrow \theta, \phi$
3: **for** $r \leftarrow 1$ to $T_r$ **do**
4:     Update $\hat{e}^{(r)}$ based on Equation (4)
5:     Update $\theta^{(r)}, \phi^{(r)}$ based on Equation (5)
6: **end for**
7: **Output:** $\theta, \phi$.

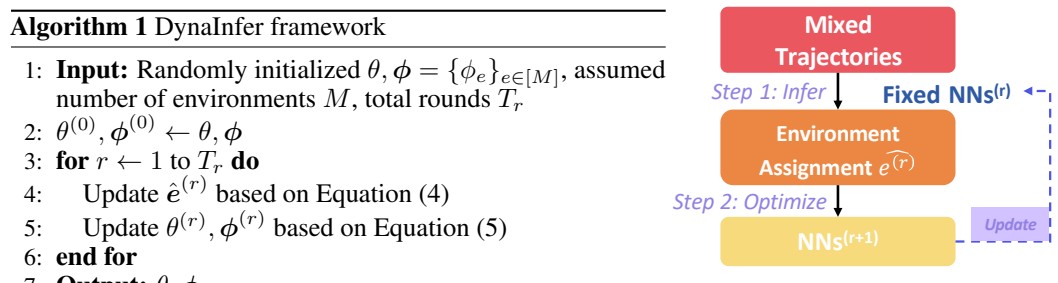

Figure 2: Model Framework.

## 3.1 Bias-aware Environment Assignment

The environment inference step receives a single dataset as input and generates a partition of the data into multiple environments. Intuitively, trajectories originating from the same environment adhere to

consistent dynamics. Employing the same neural network to model these trajectories should yield similar estimation error across them, reflecting a coherence in their dynamic parameters.

Upon examination, we observe that the optimization framework defined in Equation (2) shares conceptual similarities with classical centroid-based clustering methodologies, although the latter generally operate in Euclidean space. In K-means clustering, the primary goal is to minimize the within-cluster sum of squares, often referred to as cluster inertia [37]. This minimization effort concentrates on reducing the distances between the points within each cluster and their corresponding centroid, which typically converges to a local optimum. This characteristic enables K-means to efficiently delineate distinct and compact clusters, capturing the core essence of data distribution with respect to spatial proximity.

This insight prompts us to explore a conceptual analogy wherein the neural network that minimizes the loss most effectively operates analogously to a "centroid" for a cluster of trajectories within the same dynamic environment. We characterise the distance between a trajectory (data point) and the network (centroid) by the regression loss of the trajectory using the network. Initially, with a randomly initialized network—analogous to a randomly initialized centroid in K-means clustering—we assign each trajectory a label based on the minimal prediction loss calculated from all available networks. Subsequently, we refine this "centroid" by optimizing it to minimize the loss as specified in Equation (2). Through this iterative optimization process, we can achieve the objective stated in Equation (3).

More specifically, at round $r$, given the fixed network parameters from the previous iteration $\theta^{(r-1)}, \phi^{(r-1)}$, the environment assignment $\hat{e}_i^{(r)}$ is updated through the following process,

$$\hat{e}_i^{(r)} = \arg\min_{e \in [M]} \int_{t \in I} \left\| \frac{dx_t^i}{dt} - h\left(x_t^i; \theta^{(r-1)}, \phi_e^{(r-1)}\right) \right\|_2^2 dt. \tag{4}$$

If multiple solutions exist for Equation (4) and $\hat{e}_i^{(r-1)}$ minimizes it, we retain this assignment for the next round, i.e., $\hat{e}_i^{(r)} = \hat{e}_i^{(r-1)}$. This approach ensures the validity of a constant loss reduction (to be stated in Proposition 3.1).

## 3.2 Assignment-driven Optimization

After assigning trajectories to specific clusters, we proceed to update the conceptual centroid by optimizing network parameters. In the K-means algorithm, centroids are recalculated by averaging the positions of all points within each cluster. Similarly, our method updates network parameters by considering the mean estimation error over trajectories within a cluster, ensuring unbiased contributions from each trajectory. This approach not only improves the representational accuracy of each cluster but also enables the network to dynamically adapt to the underlying structure of the trajectories, thereby enhancing the efficacy and reliability of our learning process in unlabeled scenarios. Therefore, the parameters $\theta^{(r)}$ and $\phi^{(r)}$ are given by:

$$\theta^{(r)}, \phi^{(r)} = \arg\min_{\theta, \phi} R_{\hat{e}^{(r)}}(\theta, \phi). \tag{5}$$

## 3.3 Theoretical Property

We begin by demonstrating that Algorithm 1 effectively optimizes Equation (3).

**Proposition 3.1.** *For all rounds $1 \le r < T_r$, we must have*

$$R_{\hat{e}^{(r+1)}}\left(\theta^{(r+1)}, \phi^{(r+1)}\right) \le R_{\hat{e}^{(r)}}\left(\theta^{(r)}, \phi^{(r)}\right).$$

*Furthermore, suppose the space of $\arg\min_{\theta, \phi} R_{\hat{e}}(\theta, \phi)$ is finite for all $\hat{e} \in [M]^N$. Then there exists a constant $C > 0$ such that if $r > 1$ and $R_{\hat{e}^{(r+1)}}(\theta^{(r+1)}, \phi^{(r+1)}) < R_{\hat{e}^{(r)}}(\theta^{(r)}, \phi^{(r)})$, we must have*

$$R_{\hat{e}^{(r+1)}}\left(\theta^{(r+1)}, \phi^{(r+1)}\right) \le R_{\hat{e}^{(r)}}\left(\theta^{(r)}, \phi^{(r)}\right) - C.$$

*Remark* 3.1. Given the assumptions made in prior works [43, 17] that $h(\cdot; \theta, \phi_e)$ is linear with respect to $\theta$ and $\phi_e$, and that $\Omega(\phi_e)$ is strictly convex with respect to $\phi_e$, it follows logically that the space of $\arg\min_{\theta, \phi} R_{\hat{e}}(\theta, \phi)$ is finite for all $\hat{e} \in [M]^N$, as is evident from Equation (2). The proof is provided in Appendix A.

This proposition demonstrates that, as long as the loss in consecutive rounds of Algorithm 1 decreases, the loss must decrease by a constant $C > 0$.

# 4 Experiments

Our experiments investigate three dynamical systems governed by specific differential equations: an ODE for biological modeling, PDEs for reaction-diffusion in chemistry, and the Navier-Stokes equations for incompressible fluid dynamics. These complex, nonlinear systems test our method's ability to classify spatio-temporal patterns and physical laws across diverse environments.

## 4.1 Environment Specification

We provide a basic introduction to the datasets here, with detailed descriptions in Appendix E. **Lotka-Volterra (LV) [23]** The system models the dynamics between a prey-predator pair in an ecosystem, captured by the following ODE:

$$dm/dt = \alpha m - \beta mn, dn/dt = \delta mn - \gamma n$$

where $m$ and $n$ represent the population densities of the prey and predator, respectively, and $\alpha$, $\beta$, $\delta$, and $\gamma$ are the interaction parameters between the two species.

**Gray-Scott (GS) [28]** The model uses simple reaction-diffusion equations to effectively study complex pattern formation in chemical and biological systems, following underlying PDE dynamics:

$$\partial m/\partial t = D_m \Delta m - mn^2 + F(1 - m),$$
$$\partial n/\partial t = D_n \Delta n - mn^2 - (F + k)n.$$

where $m$ and $n$ represent the concentrations of two chemical components in the spatial domain $S$ with periodic boundary conditions; $D_m$ and $D_n$ are their constant diffusion coefficients; and $F$ and $k$ are the reaction parameters that govern the spatio-temporal dynamic patterns.

**Navier-Stokes (NS) [22]** The Navier-Stokes PDE describes the motion of viscous fluid substances:

$$\partial m/\partial t = -n\nabla m + \nu \Delta m + \xi, \nabla v = 0$$

where $n$ is the velocity field, $m = \nabla \times n$ is the vorticity, both $n$ and $m$ lie in a spatial domain $S$ with periodic boundary conditions, $\nu$ is the viscosity (fixed at $1e^{-3}$), and $\xi$ is the constant forcing term in the domain $S$.

## 4.2 Experimental Setting and Baselines

**Settings.** We evaluate DynaInfer in two distinct settings: in-domain generalization on $\mathcal{E}_o$ and adaptation to new environments in $\mathcal{E}_u$, with $\mathcal{E}_o$ and $\mathcal{E}_u$ hosting disjoint environments. For in-domain experiments, both training and testing occur on $\mathcal{E}_o$. At test time, environment labels are also not provided and are instead inferred from the prediction bias over an initial segment of the trajectory (less than $2\Delta t$ for practical reasons). For adaptation experiments, we follow standard domain adaptation practice: initial training on the source domain $\mathcal{E}_o$ is followed by fine-tuning and testing on the target domain $\mathcal{E}_u$, where environment labels are provided.

**Dataset Preparation.** For in-domain experiments, we generate four LV trajectories in each of nine environments, ten GS trajectories in each of three environments, and eight NS trajectories in each of four environments. For adaptation experiments, we simulate the same number of trajectories per environment, conducting finetuning in two additional environments $e \in \mathcal{E}_u$. All dynamic environment parameters are detailed in Appendix E. For evaluation, we sample 32 trajectories per environment, initialized according to the underlying distribution $p(x_0)$. The LV and GS data are generated using the DOPRI5 solver [8, 12], while the NS data is simulated with the pseudo-spectral method as in [22].

**Baselines.** We explore three potential strategies for assigning environment labels in the absence of environmental information, compared to our method (DynaInfer): grouping all samples into a single environment (All in One), assigning a distinct environment label to each sample (One per Env), and random assignment (Random). Additionally, we consider an "Oracle" assignment method where labels are fully known during training, bringing the total to five labeling strategies. Furthermore,

we consider three base models for dynamical system generalization: LEADS [43], CoDA-$l_1$, and CoDA-$l_2$ [17]. We utilize the neural network architectures and parameter configurations as described in their papers for each type of dynamic system. By combining these assignment methods with base models, we generate fifteen distinct methods for evaluation. In adaptation experiments, during fine-tuning, LEADS and CoDA adhere to the protocol described in their papers, by fixing the shared components or parameters and rendering only the $\mathcal{E}_u$-specific components trainable. All neural network architectures, optimizers, and parameters for the base models are configured as described in their respective papers.

**Metrics.** To rigorously evaluate predictive accuracy in dynamical system learning, we adopt two complementary metrics: Mean Squared Error (MSE) and Mean Absolute Percentage Error (MAPE), averaged over 5 independent runs.

## 4.3 Experimental Results

### 4.3.1 In-domain Generalization Results

The in-domain generalization results detailed in Table 1 illustrate the performance implications of various assignment strategies. We observe that the "All in One" and "Random" assignment strategies consistently underperform across multiple datasets and baseline models. While the "One per Env" strategy yields only mediocre results, it provides a viable initial approach in scenarios where no labels are available. Across all datasets, DynaInfer significantly outperforms other assignment strategies. Furthermore, DynaInfer consistently shows effectiveness across all tested base models and datasets, underscoring its robustness against diverse methods and datasets. Notably, DynaInfer either matches or exceeds Oracle performance, particularly in complex PDE environments like GS and NS, suggesting that its bias-aware approach effectively compensates for not having access to the true labels available to Oracle.

In Figure 3, DynaInfer's predicted states qualitatively align closely with the ground truth and Oracle, occasionally outperforming Oracle (e.g., in GS dataset with LEADS base model, where Oracle shows some jitters).

| Data | Assignment | LEADS | | | CoDA-$l_1$ | | | CoDA-$l_2$ | | |
| --- | --- | --- | --- | --- | --- | --- | --- | --- | --- | --- |
| | | Train | Test | | Train | Test | | Train | Test | |
| | | MSE | MSE | MAPE | MSE | MSE | MAPE | MSE | MSE | MAPE |
| LV | All in One | 7.17 E-2 | 7.41±0.02 E-2 | 49.22±1.84 | 7.14 E-2 | 7.40±0.01 E-2 | 49.44±3.15 | 7.17 E-2 | 7.41±0.00 E-2 | 39.26±22.13 |
| | One per Env | 4.15 E-4 | 4.91±3.50 E-4 | 6.68±2.44 | 8.68 E-4 | 9.14±0.41 E-4 | 5.67±1.01 | 8.18 E-4 | 8.43±0.39 E-4 | 5.73±1.19 |
| | Random | 7.20 E-2 | 7.38±0.02 E-2 | 50.01±1.05 | 7.12 E-2 | 7.39±0.01 E-2 | 48.87±1.81 | 7.09 E-2 | 7.39±0.00 E-2 | 48.86±2.54 |
| | **DynaInfer** | **4.74 E-5** | **7.93±2.49 E-5** | **2.83±1.62** | **9.57 E-5** | **1.83±3.40 E-4** | **3.27±2.36** | **1.71 E-4** | **1.82±3.07 E-4** | **2.02±1.66** |
| | Oracle | 4.55 E-5 | 7.02±0.76 E-5 | 1.78±0.10 | 1.78 E-5 | 3.19±0.24 E-5 | 1.26±0.06 | 1.99 E-5 | 2.72±0.18 E-5 | 1.21±0.08 |
| GS | All in One | 8.73 E-3 | 9.60±0.02 E-3 | 3008.80±892.20 | 9.24 E-3 | 9.61±0.03 E-3 | 4115.80±223.54 | 9.25 E-3 | 9.60±0.00 E-3 | 3723.00±713.85 |
| | One per Env | 1.38 E-3 | 1.65±0.54 E-3 | 173.44±59.16 | 1.56 E-3 | 1.91±0.06 E-3 | 185.23±61.43 | 1.52 E-3 | 1.87±0.02 E-3 | 174.08±57.82 |
| | Random | 8.78 E-3 | 9.36±0.20 E-3 | 1403.50±119.50 | 9.25 E-3 | 9.59±0.03 E-3 | 3958.25±682.38 | 8.77 E-3 | 9.35±0.02 E-3 | 3919.88±157.54 |
| | **DynaInfer** | **3.60 E-5** | **4.14±0.21 E-5** | **117.57±33.90** | **9.22 E-5** | **1.23±0.41 E-4** | **122.93±22.05** | **6.69 E-5** | **7.25±2.11 E-5** | **112.52±14.15** |
| | Oracle | 7.73 E-5 | 1.34±0.76 E-4 | 97.77±12.09 | 6.04 E-5 | 9.60±3.91 E-5 | 163.38±47.89 | 4.69 E-5 | 7.04±1.84 E-5 | 138.86±16.55 |
| NS | All in One | 5.34E-02 | 6.71±0.11 E-2 | 239.70±14.78 | 5.79E-02 | 6.64±0.11 E-2 | 251.38±8.54 | 6.17E-02 | 6.64±0.03 E-2 | 255.26±9.11 |
| | One per Env | 2.24E-02 | 4.11±0.14 E-2 | 169.48±9.68 | 3.45E-02 | 3.88±0.22 E-2 | 161.04±10.73 | 2.31E-02 | 4.04±0.22 E-2 | 158.26±9.35 |
| | Random | 3.06E-02 | 6.58±0.05 E-2 | 233.80 ± 8.44 | 5.04E-02 | 6.58±0.05 E-2 | 247.95±4.59 | 5.78E-02 | 6.66±0.04 E-2 | 254.47±6.40 |
| | **DynaInfer** | **6.10E-04** | **7.05±0.34 E-3** | **77.29±10.18** | **1.23E-02** | **1.62±0.18 E-2** | **108.17±10.30** | **8.92E-04** | **1.19±0.12 E-2** | **96.57±12.75** |
| | Oracle | 2.59E-04 | 6.55±1.34 E-3 | 67.58±9.37 | 1.36E-02 | 1.73±0.29 E-2 | 124.22±12.35 | 7.11E-04 | 9.46±0.51 E-3 | 91.06±5.85 |

Table 1: In-domain Experiment Results on the LV, GS, and NS environments. Our approach consistently outperforms all non-oracle assignment methods, and beats oracle at times, demonstrating its effectiveness in modeling heterogeneous environments and generalizing across dynamical systems.

### 4.3.2 Domain Adaptation Results

The results in Table 2 demonstrate the performance of different assignment strategies under the domain adaptation setting. The "One per Env" strategy consistently outperforms the "All in One" approach. While the "Random" assignment benefits the base model LEADS, it slightly diminishes CoDA's performance across all datasets. DynaInfer shows strong adaptation capabilities across various datasets and base generalization methods, consistently outperforming other non-Oracle techniques. This indicates that DynaInfer effectively captures commonalities across environments, enabling smoother adaptation to new conditions. Furthermore, the performance gap between DynaInfer and the Oracle is significantly narrower in adaptation tasks compared to in-domain generalization.

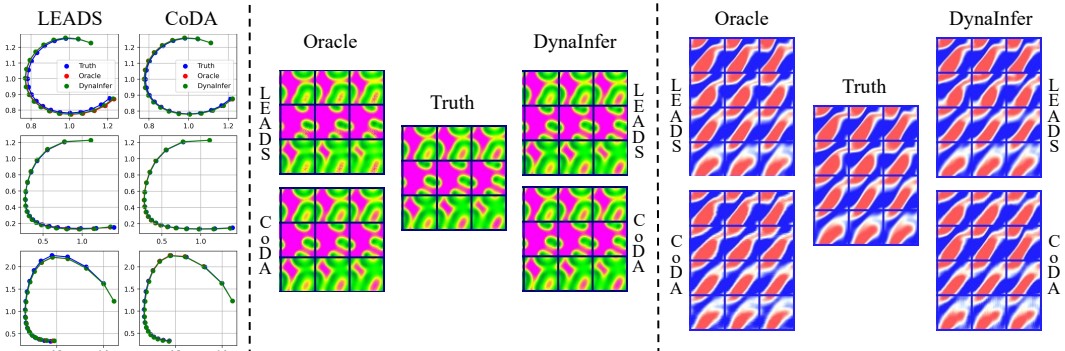

Figure 3: Left: Predicted test trajectories from 3 environments vs. ground truth and Oracle for LV with 2 base models. Middle and Right: Predicted last 3 states for GS and NS, respectively, vs. ground truth and Oracle using 2 base models. Each environment is shown by row. For CoDA, we use the best Oracle result's CoDA version to save space. See Appendix H for all the visualizations.

| Data | Assignment | LEADS | | CoDA-$l_1$ | | CoDA-$l_2$ | |
|---|---|---|---|---|---|---|---|
| | | MSE | MAPE | MSE | MAPE | MSE | MAPE |
| LV | All in One | 4.16±8.61 E-2 | 9.92±3.55 | 4.01±6.43 E-2 | 26.90±10.65 | 4.10±7.61 E-2 | 27.80±8.81 |
| | One per Env | 2.28±1.81 E-3 | 5.41±0.50 | 1.72±0.53 E-3 | 27.63±7.81 | 1.66±0.82 E-3 | 25.51±7.72 |
| | Random | 1.72±0.53 E-3 | 6.87±0.13 | 1.14±0.61 E-3 | 27.56±6.36 | 1.05±0.54 E-3 | 29.65±6.31 |
| | DynaInfer | **5.77±1.46 E-4** | **2.84±0.13** | **8.37±0.94 E-5** | **10.16±0.04** | **8.49±2.07 E-5** | **10.30±0.08** |
| | Oracle | 1.67±2.26 E-3 | 3.16±0.37 | 5.85±1.24 E-5 | 10.24±0.06 | 5.12±3.17 E-5 | 10.24±0.04 |
| GS | All in One | 4.59±1.18 E-4 | 721.25±197.54 | 2.85±0.55 E-3 | 6658.20±1651.77 | 2.76±0.72 E-3 | 7355.20±335.45 |
| | One per Env | 3.59±2.71 E-4 | 450.20±368.36 | 1.08±0.82 E-3 | 6247.34±817.74 | 1.19±0.97 E-3 | 5948.57±935.71 |
| | Random | 5.73±0.86 E-4 | 1261.00±979.30 | 2.92±0.80 E-3 | 7292.88±312.54 | 2.84±0.89 E-3 | 4499.25±844.62 |
| | DynaInfer | **1.00±0.32 E-4** | **378.73±182.12** | **2.41±0.91 E-4** | **220.54±65.60** | **2.13±0.41 E-4** | **207.96±46.49** |
| | Oracle | 2.21±0.93 E-4 | 434.73±432.38 | 2.66±0.79 E-4 | 302.68±188.25 | 2.10±0.89 E-4 | 230.62±118.29 |
| NS | All in One | 1.25±2.04 E-2 | 67.17±3.33 | 1.25±0.20 E-2 | 218.66±27.26 | 1.29±0.29 E-2 | 214.44±17.08 |
| | One per Env | 2.78±2.08 E-2 | 96.23±4.54 | 2.04±0.68 E-2 | 214.38±15.19 | 2.43±0.48 E-2 | 209.68±18.98 |
| | Random | 1.32±0.53 E-2 | 81.18±7.43 | 4.66±1.04 E-2 | 215.21±19.39 | 4.37±0.99 E-2 | 191.03±16.38 |
| | DynaInfer | **7.52±0.76 E-3** | **50.93±8.83** | **9.27±1.81 E-3** | **101.38±15.77** | **9.71±2.10 E-3** | **101.04±16.27** |
| | Oracle | 1.16±0.68 E-2 | 57.35±17.90 | 7.46±0.72 E-3 | 154.86±41.00 | 7.32±0.81 E-3 | 100.04±24.93 |

Table 2: Adaptation Experiment Results on the LV, GS, and NS environments. DynaInfer consistently outperforms other non-Oracle methods across all datasets and narrows the performance gap with the Oracle more effectively compared to in-domain generalization.

### 4.3.3 Assignments Convergence

We illustrate the probability of environment assignments with DynaInfer over training time in Figure 4. Initially, our model may default to random assignments due to unoptimized neural networks. However, the assignments quickly converge to the true labels. Notably, systems with simpler dynamics, like LV compared to NS, enable quicker learning of base generalization methods, resulting in faster convergence of environment assignments.

### 4.3.4 Performance across Varying Number of Assumed Environments

As the true number of environments ($|\mathcal{E}_o|$) might be unknown, we examine DynaInfer's performance with varying assumed environments $M$ in Figure 5. Our findings show that prior knowledge of the true $M$ is beneficial: performance peaks when the assumed $M$ aligns with the true count. Additionally, our model demonstrates robustness to over-estimations of $M$. Due to its ability to account for bias, our model effectively identifies trajectories from the same environment and remains robust to an excess of inaccurately trained neural networks, even when the assumed $M$ is too large. The above observations suggest that incrementally increasing the number of environments until peak performance is achieved can be a straightforward way to identify the true $M$. Lastly, DynaInfer consistently outperforms other non-oracle approaches when $M$ is underestimated, except for the One-per-Env baseline (which requires $M$ equal to the number of trajectories and is computationally infeasible).

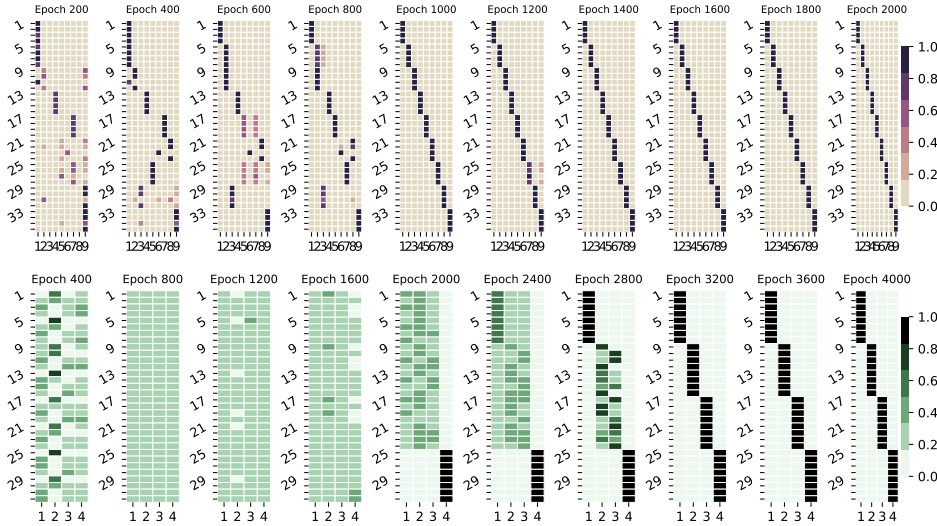

Figure 4: Environment assignment probability over time, averaged over 5 runs, with LEADS as base model (on LV (top) and NS (bottom); see Appendix G for GS). The assignment converges to the true label faster than the designated training steps. A similar trend is observed with the CoDA model.

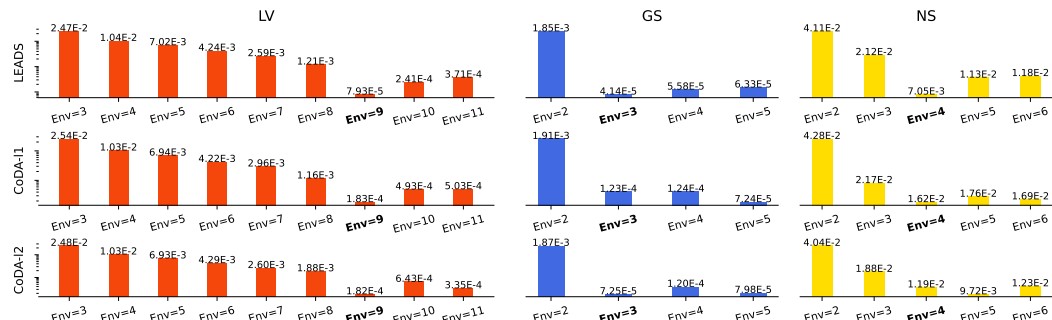

Figure 5: Performance across assumed number of environments with DynaInfer(log-scale y-axis). The peak performance aligns with the true number of environments (bold on x-axis) with high probability, and remains stable thereafter.

# 5 Related Work

## 5.1 Domain Generalization and Adaptation

Domain generalization (DG) seeks to train a model on one or multiple distinct but related source domains so that it generalizes effectively to any out-of-distribution (OOD) target domain. DG methods assume data heterogeneity and use additional environment labels to develop models that remain robust across unseen and shifted test data. Many DG strategies focus on domain alignment, aiming to minimize divergence among source domains to achieve domain-invariant representations [21, 14, 29, 31, 40]. Other approaches enhance the diversity of training data by augmenting source domains [5, 33, 39]. Additionally, some methods leverage meta-learning and invariant risk minimization for regularization, further enhancing generalization [20, 1].

Domain adaptation (DA) methods enable model generalization to target domains with shifted data distributions and are primarily classified into three categories. Instance-based methods reweight or adjust training samples to reflect the test distribution [15, 6]. Feature-based approaches align feature distributions across training and test domains [38, 36]. Model-based strategies focus on developing models that are either robust to domain shifts or specifically tailored for the target domain [25, 10].

### 5.2 Generalization for Dynamical Systems

Generalization in dynamical systems remains underexplored in literature. Among the limited studies, LEADS emerges as a novel multi-task learning framework that effectively generalizes across the functional space of dynamical systems [43]. Alternatively, CoDA optimizes within the parameter space, enhancing model adaptability and efficiency while accommodating increased environmental variability without requiring multiple distinct network trainings for each setting [17]. In contrast, DyAd is a context-aware meta-learning approach that adjusts the dynamics model by decoding a time-invariant context from observed states [41]. Despite its novelty, DyAd relies on potentially impractical weak supervision based on physics-derived quantities and uses Adaptive Instance Normalization, which may degrade performance.

Currently, three notable weaknesses prevail in generalization works for dynamical systems. First, there is an assumption that prior knowledge about the target domain exists, and without it, most generalization methods would fail [11]. Second, the predominant use of the mean squared error as a loss function is inadequate for evaluating the reconstruction accuracy of chaotic systems. Lastly, the influence of unlabelled trajectory data on the process of learning generalizable dynamical systems remains both unexplored and unresolved — a gap this paper examines for the first time. While switched systems learning methods [19] infer modes by classifying individual data points (analogous to environment inference), our work operates on trajectories governed by ODEs or PDEs.

## 6 Conclusion

We propose an environment inference method that improves the understanding and generalization of complex dynamical systems across various environments without using manually labeled data. DynaInfer infers environment labels directly from training data, overcoming the challenges associated with explicit annotation. Theoretical analysis ensures convergence of DynaInfer, and experiments show DynaInfer often surpasses non-oracle methods and matches or exceeds oracle performance.

Future research could unfold along the following promising directions: First, to improve generalization in chaotic systems, MSE-based methods could be replaced with more suitable metrics such as the sliced Wasserstein-1 distance, which would require developing a tailored inference model. Similarly, effectively inferring environments for dynamics with complex boundaries remains a significant open problem, as current learning methods often oversimplify boundary conditions. Furthermore, for systems requiring interpretable parameters, our method could be extended to jointly optimize environment labels and physical coefficients. To improve convergence, coordinate optimization techniques may help escape local optima for objectives that are convex in individual variable blocks [13, 30]. Finally, techniques such as adaptive early stopping, dynamic batching, and membership functions [3] could further enhance training efficiency.

### Acknowledgments and Disclosure of Funding

This work was supported by NSFC (No. 62425206, 62141607, 62525213), Beijing Municipal Science and Technology Project (No. Z241100004224009), and Big Data and Responsible Artificial Intelligence for National Governance, Renmin University of China. We also extend our thanks to Renzhe Xu and Hao Zou for their insightful discussions.

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

# A  Proof of Proposition 3.1

*Proof.* Due to the operation in Equation (5), we must have

$$R_{\hat{e}^{(r+1)}}\left(\theta^{(r+1)}, \phi^{(r+1)}\right) \leq R_{\hat{e}^{(r+1)}}\left(\theta^{(r)}, \phi^{(r)}\right).$$

In addition, according to the operation in Equation (4), we must have

$$\forall i \in [N], \int_{t \in I} \left\| \frac{dx_t^i}{dt} - h\left(x_t^i; \theta^{(r)}, \phi_{e_i^{(r+1)}}^{(r)}\right) \right\|_2^2 dt$$

$$\leq \int_{t \in I} \left\| \frac{dx_t^i}{dt} - h\left(x_t^i; \theta^{(r)}, \phi_{e_i^{(r)}}^{(r)}\right) \right\|_2^2 dt.$$

As a result, since the regularization terms (*i.e.*, the $\Omega(\phi_e)$ term) in $R_{e^{\hat{r}+1}}(\theta^{(r)}, \phi^{(r)})$ and $R_{\hat{e^r}}(\theta^{(r)}, \phi^{(r)})$ remain the same, we must have

$$R_{\hat{e}^{(r+1)}}\left(\theta^{(r)}, \phi^{(r)}\right) \leq R_{\hat{e}^{(r)}}\left(\theta^{(r)}, \phi^{(r)}\right).$$

Now we have

$$R_{\hat{e}^{(r+1)}}\left(\theta^{(r+1)}, \phi^{(r+1)}\right) \leq R_{\hat{e}^{(r+1)}}\left(\theta^{(r)}, \phi^{(r)}\right) \leq R_{\hat{e}^{(r)}}\left(\theta^{(r)}, \phi^{(r)}\right).$$

Now consider the second part of the proposition. Define the following space of $\theta, \phi$

$$\mathcal{H} \triangleq \left\{ (\theta, \phi) : \exists \hat{e} \in [M]^N \text{ such that } R_{\hat{e}}(\theta, \phi) = \min_{\theta', \phi'} R_{\hat{e}}(\theta', \phi') \right\}.$$

Note that by the assumption, we must have $|\mathcal{H}| < \infty$. Now define $\mathcal{A}$ as

$$\mathcal{A} = \left\{ \int_{t \in I} \left\| \frac{dx_t^i}{dt} - h\left(x_t^i; \theta, \phi_e\right) \right\|_2^2 dt : i \in [N], e \in [M], (\theta, \phi) \in \mathcal{H} \right\}.$$

Note that since $|\mathcal{H}| < \infty$, we have $|\mathcal{A}| < \infty$. $C$ is defined as

$$C = \min_{a, b \in \mathcal{A}, a \neq b} |a - b|.$$

Since $|\mathcal{A}| < \infty$, we must have $C > 0$. In addition, note that when $r > 1$ and $R_{\hat{e}^{(r+1)}}(\theta^{(r+1)}, \phi^{(r+1)}) < R_{\hat{e}^{(r)}}(\theta^{(r)}, \phi^{(r)})$, we must have $\hat{e}^{(r+1)} \neq \hat{e}^{(r)}$. Otherwise we will have

$$R_{\hat{e}^{(r+1)}}\left(\theta^{(r+1)}, \phi^{(r+1)}\right) = R_{\hat{e}^{(r)}}\left(\theta^{(r+1)}, \phi^{(r+1)}\right)$$

$$= R_{\hat{e}^{(r)}}\left(\theta^{(r)}, \phi^{(r)}\right).$$

As a result, there exists $i \in [N]$ such that $\hat{e}_i^{(r+1)} \neq \hat{e}_i^{(r)}$. By the choice of $\hat{e}_i^{(r+1)}$, we must have

$$\int_{t \in I} \left\| \frac{dx_t^i}{dt} - h\left(x_t^i; \theta^{(r)}, \phi_{\hat{e}_i^{(r)}}^{(r)}\right) \right\|_2^2 dt$$

$$\neq \int_{t \in I} \left\| \frac{dx_t^i}{dt} - h\left(x_t^i; \theta^{(r)}, \phi_{\hat{e}_i^{(r+1)}}^{(r)}\right) \right\|_2^2 dt.$$

As a result, by the definition of $C$, we have

$$\int_{t \in I} \left\| \frac{dx_t^i}{dt} - h\left(x_t^i; \theta^{(r)}, \phi_{\hat{e}_i^{(r)}}^{(r)}\right) \right\|_2^2 dt$$

$$- \int_{t \in I} \left\| \frac{dx_t^i}{dt} - h\left(x_t^i; \theta^{(r)}, \phi_{\hat{e}_i^{(r+1)}}^{(r)}\right) \right\|_2^2 dt \geq C.$$

Therefore, we must have

$$R_{\hat{e}^{(r+1)}}\left(\theta^{(r+1)}, \phi^{(r+1)}\right) \leq R_{\hat{e}^{(r)}}\left(\theta^{(r)}, \phi^{(r)}\right) - C.$$

Now the claim follows. $\qquad \square$

## B Real-world Experiment

We further evaluated our method using a real-world robot motion trajectory dataset [16]. This dataset comprises three distinct motion patterns: (1) drawing "S" shapes, (2) placing a cube on a shelf, and (3) drawing large "C" shapes. In our experimental framework, each of these patterns is treated as a distinct environment. The objective of this evaluation is to assess whether our method can accurately infer the underlying environment, thereby supporting the learning of a generalizable neural network for simulating their dynamics. To ensure a consistent input structure across environments, all trajectories were projected into a two-dimensional space. The dataset was partitioned into training and test sets with a ratio of 6:1. Furthermore, each trajectory was temporally resampled to a standardized length of 100 time steps for smoothing and uniformity following the practice in [44].

The empirical results are presented in Table 3. In summary, these findings collectively suggest that our approach maintains its efficacy on real-world data, thereby substantiating its practical utility and robustness.

| Data | Assignment | LEADS | | | CoDA-$l_1$ | | | CoDA-$l_2$ | | |
| | | Train | Test | | Train | Test | | Train | Test | |
| | | MSE | MSE | MAPE | MSE | MSE | MAPE | MSE | MSE | MAPE |
|---|---|---|---|---|---|---|---|---|---|---|
| RM | All in One | 7.17 E-2 | 7.41±0.02 E-2 | 49.22±1.84 | 7.14 E-2 | 7.40±0.01 E-2 | 49.44±3.15 | 7.17 E-2 | 7.41±0.00 E-2 | 39.26±22.13 |
| | One per Env | 4.15 E-4 | 4.91±3.50 E-4 | 6.68±2.44 | 8.68 E-4 | 9.14±0.41 E-4 | 5.67±1.01 | 8.18 E-4 | 8.43±0.39 E-4 | 5.73±1.19 |
| | Random | 7.20 E-2 | 7.38±0.02 E-2 | 50.01±1.05 | 7.12 E-2 | 7.39±0.01 E-2 | 48.87±1.81 | 7.09 E-2 | 7.39±0.00 E-2 | 48.86±2.54 |
| | DynaInfer | **4.74 E-5** | **7.93±2.49 E-5** | **2.83±1.62** | 9.57 E-5 | 1.83±3.40 E-4 | 3.27±2.36 | 1.71 E-4 | 1.82±3.07 E-4 | 2.02±1.66 |
| | Oracle | 4.55 E-5 | 7.02±0.76 E-5 | 1.78±0.10 | 1.78 E-5 | 3.19±0.24 E-5 | 1.26±0.06 | 1.99 E-5 | 2.72±0.18 E-5 | 1.21±0.08 |

Table 3: In-domain Experiment Results on the Robot Motion.

## C Sensitivity Analysis to $|\mathcal{E}_o|$

To assess the robustness of our method to the number of underlying environments, we conducted a sensitivity analysis using the LV system. We systematically varied the true number of environments, $|\mathcal{E}_o|$, from 2 to 16 and evaluated the performance of our model against the Oracle baseline that has privileged access to the true environment labels. The results are presented in Table 4. DynaInfer, achieves performance comparable to the Oracle across all tested values of $|\mathcal{E}_o|$. This empirical evidence demonstrates that our approach is robust to variations in the quantity of environments.

Table 4: Test Mean Squared Error on the LV system for varying $|\mathcal{E}_o|$

| | $\mathcal{E}_o$ | | | | | | |
| | 2 | 3 | 4 | 5 | 6 | 7 | 8 |
|---|---|---|---|---|---|---|---|
| DynaInfer | 2.12E-5 | 4.22E-5 | 6.77E-5 | 6.24E-5 | 8.81E-5 | 6.70E-5 | 6.54E-5 |
| Oracle | 2.76E-5 | 3.99E-5 | 5.75E-5 | 5.20E-5 | 7.24E-5 | 8.14E-5 | 5.74E-5 |

| | $\mathcal{E}_o$ | | | | | | | |
| | 9 | 10 | 11 | 12 | 13 | 14 | 15 | 16 |
|---|---|---|---|---|---|---|---|---|
| DynaInfer | 6.33E-5 | 8.34E-5 | 2.24E-4 | 1.34E-4 | 9.69E-5 | 1.97E-4 | 3.43E-4 | 2.73E-4 |
| Oracle | 7.60E-5 | 9.31E-5 | 1.85E-4 | 1.23E-4 | 1.21E-4 | 1.71E-4 | 2.56E-4 | 2.48E-4 |

## D Robustness over $M$

We assessed our robustness to the overestimation of $M$, the assumed number of environments, in the LV system. The system's true value is $M^* = 9$, with each environment containing four trajectories. In our experiments, we varied the assumed $M$ from 5 to 36. The results, shown in Table 5, demonstrate that DynaInfer effectively adapts to this overestimation. The learned label count consistently converged to a range of 8–11, which aligns closely with the ground-truth value.

Furthermore, we conducted a comprehensive analysis by sweeping $M$ across its full spectrum, from $M = 1$ to $M = $ #trajectory. As shown in Table 6, the optimal performance was achieved at $M = 9$ and the model maintained robustness even when $M$ exceeded this value.

Table 5: Learned Label Counts for different values of $M$ averaged over 5 runs

| M | 5 | 6 | 7 | 8 | 9 | 10 | 11 | 12 | 13 | 14 | 15 | 16 | 17 | 18 | 19 | 20 |
|---|---|---|---|---|---|---|---|---|---|---|---|---|---|---|---|---|
| # Labels | 5 | 6 | 7 | 8 | 9 | 8.8 | 9 | 10.2 | 9 | 9 | 10 | 10.8 | 10 | 11 | 11 | 10 |

| M | 21 | 22 | 23 | 24 | 25 | 26 | 27 | 28 | 29 | 30 | 31 | 32 | 33 | 34 | 35 | 36 |
|---|---|---|---|---|---|---|---|---|---|---|---|---|---|---|---|---|
| # Labels | 10.8 | 11 | 9 | 10 | 10.2 | 11 | 10 | 10 | 10 | 11.2 | 10 | 9.2 | 10 | 10.8 | 10 | 11 |

Table 6: Test MSE over the full spectrum of $M$ for the LV system

| $M$ | MSE | $M$ | MSE | $M$ | MSE | $M$ | MSE |
|---|---|---|---|---|---|---|---|
| 1 | 7.41E-2 | 10 | 2.41E-4 | 19 | 2.60E-4 | 28 | 2.04E-4 |
| 2 | 4.01E-2 | 11 | 3.71E-4 | 20 | 1.36E-4 | 29 | 3.65E-4 |
| 3 | 2.47E-2 | 12 | 2.57E-4 | 21 | 1.21E-4 | 30 | 1.42E-3 |
| 4 | 1.04E-2 | 13 | 1.63E-4 | 22 | 2.45E-4 | 31 | 3.84E-4 |
| 5 | 7.02E-3 | 14 | 2.24E-4 | 23 | 9.06E-5 | 32 | 1.16E-4 |
| 6 | 4.24E-3 | 15 | 1.15E-4 | 24 | 2.38E-4 | 33 | 3.89E-4 |
| 7 | 2.59E-3 | 16 | 1.76E-4 | 25 | 8.27E-5 | 34 | 3.10E-4 |
| 8 | 1.21E-3 | 17 | 1.83E-4 | 26 | 1.06E-4 | 35 | 1.72E-4 |
| 9 | 7.93E-5 | 18 | 3.60E-4 | 27 | 3.71E-4 | 36 | 2.77E-4 |

# E  Environment Specification

We conducted experiments on a server equipped with a 64-core CPU, 256 GB of RAM, and eight 24GB RTX-3090Ti GPUs. The DynaInfer framework was implemented using PyTorch [27]. All NN params in our method are randomly initialized.

**Lotka-Volterra (LV) [23]**   The system models the dynamics between a prey-predator pair in an ecosystem, captured by the following ODE:

$$dm/dt = \alpha m - \beta mn, dn/dt = \delta mn - \gamma n$$

, where $m, n$ represent the population density of the prey and predator, respectively, and $\alpha, \beta, \delta, \gamma$ are the interaction parameters between the two species. The system state is defined as $x_t^e = (m_t^e, n_t^e) \in \mathbb{R}_+^2$ with initial conditions $(m_0, n_0)$ sampled from a uniform distribution $p(x_0) = Unif([1,3]^2)$. The environment $e$ is defined by dynamics parameters $\theta_e = (\alpha_e/\beta_e, \gamma_e/\delta_e) \in \Theta$, sampled uniformly from the set $\Theta$. We simulate trajectories over a temporal grid with $\Delta t = 0.5$ and a horizon $T = 10$. At test time, environment labels are inferred from the prediction bias observed over an initial segment of the trajectory of length $\Delta t$.

**Gray-Scott (GS) [28]**   The model uses simple reaction-diffusion equations to effectively study complex pattern formation in chemical and biological systems, following underlying PDE dynamics:

$$\partial m/\partial t = D_m \Delta m - mn^2 + F(1-m),$$
$$\partial n/\partial t = D_n \Delta n - mn^2 - (F+k)n.$$

, where $m, n$ represent the concentrations of two chemical components in the spatial domain $S$ with periodic boundary conditions, and $D_m, D_n$ are their constant diffusion coefficients, and $F, k$ are the reaction parameters that govern the spatio-temporal dynamic patterns. $S$ is a 2D space of dimension $32 \times 32$ with spatial resolution of $\Delta s = 2$. The system state $x_t^e = (m_t^e, n_t^e) \in \mathbb{R}_+^{2 \times 32^2}$. We define the initial conditions $(m_0, n_0) \sim p(x_0)$ by uniformly sampling three two-by-two squares, which activate the reactions, from $S$. $(m_0, n_0) = (1 - \epsilon, \epsilon)$ with $\epsilon = 0.05$ inside the squares and $(m_0, n_0) = (0, 1)$ outside the squares. The environment $e$ is defined by dynamics parameters $\theta = (F_e, k_e) \in \Theta$, sampled uniformly from the environment distribution $Q$ on $\Theta$. We simulate trajectories on a temporal grid using a timestep of $\Delta t = 40$ over a horizon of $T = 400$. At test time, environment labels are inferred from an initial segment of length $\Delta t$.

**Navier-Stokes (NS) [22]**   The Navier-Stokes PDE describes the motion of viscous fluid substances:

$$\partial m/\partial t = -n\nabla m + \nu \Delta m + \xi, \nabla v = 0$$

, where $n$ is the velocity field, $m = \nabla \times n$ is the vorticity, both $n, m$ lie in a spatial domain $S$ with periodic boundary conditions, $\nu$ is the viscosity (fixed as $1e^{-3}$) and $\xi$ is the constant forcing term in the domain $S$. The system state is characterized by $x_t^e = m_t^e \in \mathbb{R}^{32^2}$, as initialized per [22]. The environment $e$ is determined by a uniformly sampled forcing term $\xi_e \in \Theta_\xi$. We simulate trajectories across a temporal interval $\Delta t = 1$ over a horizon $T = 10$. At test time, environment labels are inferred from an initial segment of length $2\Delta t$.

The parameters for LV, GS and NS systems are respectively given in Table 7, 8 and 9.

Table 7: Parameters of LV Systems

| Params. | Train 1 | Train 2 | Train 3 | Train 4 | Train 5 | Train 6 | Train 7 | Train 8 | Train 9 | Adapt 1 | Adapt 2 |
|---|---|---|---|---|---|---|---|---|---|---|---|
| $\alpha$ | 0.5 | 0.5 | 0.5 | 0.5 | 0.5 | 0.5 | 0.5 | 0.5 | 0.5 | 0.7 | 0.6 |
| $\beta$ | 0.5 | 0.75 | 1 | 0.5 | 0.5 | 0.75 | 0.75 | 1 | 1 | 0.8 | 0.7 |
| $\gamma$ | 0.5 | 0.5 | 0.5 | 0.5 | 0.5 | 0.5 | 0.5 | 0.5 | 0.5 | 0.5 | 0.5 |
| $\delta$ | 0.5 | 0.5 | 0.5 | 0.75 | 1 | 0.75 | 1 | 0.75 | 1 | 0.5 | 0.5 |

Table 8: Parameters of GS Systems

| Params. | Train 1 | Train 2 | Train 3 | Adapt 1 | Adapt 2 |
|---|---|---|---|---|---|
| $F$ | 0.037 | 0.03 | 0.039 | 0.033 | 0.036 |
| $k$ | 0.06 | 0.062 | 0.058 | 0.059 | 0.061 |

Table 9: Parameters of NS Systems

| | $\xi$ |
|---|---|
| Train 1 | $0.1 * (\sin(2\pi(X + Y)) + \cos(2\pi * (X + Y)))$ |
| Train 2 | $0.1 * (\sin(2\pi(X + Y)) + \cos(2\pi * (X + 2Y)))$ |
| Train 3 | $0.1 * (\sin(2\pi(X + Y)) + \cos(2\pi * (2X + Y)))$ |
| Train 4 | $0.1 * (\sin(2\pi(X + 2Y)) + \cos(2\pi * (2X + Y)))$ |
| Adapt 1 | $0.1 * (\sin(2\pi(2X + Y)) + \cos(2\pi * (X + 2Y)))$ |
| Adapt 2 | $0.1 * (\sin(2\pi(2X + Y)) + \cos(2\pi * (2X + Y)))$ |

# F    Centroid-like NN Behavior

We conducted an additional experiment measuring the MSE loss for each training sample across different neural networks (NNs). The results, shown in Table 10, yielded two key findings: first, each training sample has a uniquely best-fit NN where it attains minimal loss. Second, this optimal NN generalizes effectively to a broader cluster of trajectories—specifically, those originating from the same environment. This alignment between a "centroid-like" NN and environment-specific trajectory clusters strongly reinforces our methodological rationale.

| Traj. | NN1 | NN2 | NN3 | NN4 | NN5 | NN6 | NN7 | NN8 | NN9 | Assign. |
|---|---|---|---|---|---|---|---|---|---|---|
| 1 | 2.06e-01 | 1.18e-01 | 2.80e-01 | 7.76e-06 | 8.34e-02 | 1.59e-01 | 3.14e-01 | 2.50e-01 | 2.35e-01 | 3 |
| 2 | 2.05e-01 | 1.21e-01 | 2.66e-01 | 9.35e-06 | 8.42e-02 | 1.53e-01 | 3.36e-01 | 2.40e-01 | 2.26e-01 | 3 |
| 3 | 2.05e-01 | 1.21e-01 | 2.65e-01 | 1.03e-05 | 8.41e-02 | 1.52e-01 | 3.35e-01 | 2.39e-01 | 2.25e-01 | 3 |
| 4 | 2.45e-01 | 1.51e-01 | 3.85e-01 | 1.98e-05 | 1.13e-01 | 1.89e-01 | 4.05e-01 | 3.10e-01 | 3.12e-01 | 3 |
| 5 | 3.31e-02 | 1.65e-01 | 1.12e-01 | 8.32e-02 | 5.36e-06 | 5.42e-02 | 3.68e-01 | 7.26e-02 | 1.37e-01 | 4 |
| 6 | 3.15e-02 | 1.87e-01 | 1.07e-01 | 8.46e-02 | 4.26e-06 | 5.57e-02 | 4.21e-01 | 6.80e-02 | 1.44e-01 | 4 |
| 7 | 3.14e-02 | 1.86e-01 | 1.06e-01 | 8.46e-02 | 4.50e-06 | 5.52e-02 | 4.19e-01 | 6.74e-02 | 1.43e-01 | 4 |
| 8 | 2.84e-02 | 2.68e-01 | 2.13e-01 | 1.14e-01 | 2.22e-05 | 9.57e-02 | 5.38e-01 | 1.18e-01 | 2.36e-01 | 4 |
| 9 | 7.60e-06 | 2.32e-01 | 5.48e-02 | 2.07e-01 | 3.36e-02 | 4.44e-02 | 4.22e-01 | 1.91e-02 | 1.18e-01 | 0 |
| 10 | 6.45e-06 | 2.64e-01 | 6.07e-02 | 2.05e-01 | 3.12e-02 | 5.17e-02 | 4.93e-01 | 2.10e-02 | 1.37e-01 | 0 |
| 11 | 6.65e-06 | 2.63e-01 | 6.02e-02 | 2.05e-01 | 3.11e-02 | 5.14e-02 | 4.91e-01 | 2.07e-02 | 1.36e-01 | 0 |
| 12 | 1.95e-05 | 3.66e-01 | 1.61e-01 | 2.45e-01 | 2.92e-02 | 9.83e-02 | 6.31e-01 | 7.05e-02 | 2.32e-01 | 0 |
| 13 | 2.31e-01 | 1.35e-05 | 1.56e-01 | 1.16e-01 | 1.64e-01 | 8.52e-02 | 5.28e-02 | 1.84e-01 | 7.04e-02 | 1 |
| 14 | 2.63e-01 | 1.39e-05 | 1.64e-01 | 1.22e-01 | 1.87e-01 | 9.48e-02 | 5.95e-02 | 2.02e-01 | 7.16e-02 | 1 |
| 15 | 2.62e-01 | 1.41e-05 | 1.63e-01 | 1.22e-01 | 1.86e-01 | 9.44e-02 | 5.93e-02 | 2.01e-01 | 7.13e-02 | 1 |
| 16 | 3.65e-01 | 1.89e-05 | 2.33e-01 | 1.50e-01 | 2.67e-01 | 1.19e-01 | 6.96e-02 | 2.56e-01 | 1.08e-01 | 1 |
| 17 | 4.22e-01 | 5.35e-02 | 2.43e-01 | 3.15e-01 | 3.68e-01 | 1.95e-01 | 2.78e-05 | 3.17e-01 | 1.13e-01 | 6 |
| 18 | 4.91e-01 | 5.91e-02 | 2.77e-01 | 3.35e-01 | 4.19e-01 | 2.27e-01 | 2.12e-05 | 3.66e-01 | 1.30e-01 | 6 |
| 19 | 4.89e-01 | 5.89e-02 | 2.76e-01 | 3.34e-01 | 4.17e-01 | 2.26e-01 | 2.08e-05 | 3.64e-01 | 1.30e-01 | 6 |
| 20 | 6.28e-01 | 6.90e-02 | 3.26e-01 | 4.04e-01 | 5.35e-01 | 2.56e-01 | 1.99e-05 | 4.18e-01 | 1.52e-01 | 6 |
| 21 | 4.36e-02 | 8.60e-02 | 1.90e-02 | 1.58e-01 | 5.31e-02 | 7.09e-06 | 1.96e-01 | 1.98e-02 | 2.19e-02 | 5 |
| 22 | 5.16e-02 | 9.53e-02 | 1.74e-02 | 1.54e-01 | 5.58e-02 | 8.08e-06 | 2.28e-01 | 2.09e-02 | 2.34e-02 | 5 |
| 23 | 5.13e-02 | 9.49e-02 | 1.73e-02 | 1.53e-01 | 5.53e-02 | 7.86e-06 | 2.27e-01 | 2.08e-02 | 2.33e-02 | 5 |
| 24 | 9.81e-02 | 1.19e-01 | 3.84e-02 | 1.89e-01 | 9.61e-02 | 1.45e-05 | 2.56e-01 | 2.72e-02 | 3.41e-02 | 5 |
| 25 | 1.17e-01 | 7.14e-02 | 2.46e-02 | 2.35e-01 | 1.36e-01 | 2.11e-02 | 1.15e-01 | 5.47e-02 | 8.69e-06 | 8 |
| 26 | 1.37e-01 | 7.21e-02 | 2.82e-02 | 2.26e-01 | 1.44e-01 | 2.36e-02 | 1.31e-01 | 6.42e-02 | 5.20e-06 | 8 |
| 27 | 1.36e-01 | 7.18e-02 | 2.81e-02 | 2.26e-01 | 1.43e-01 | 2.35e-02 | 1.31e-01 | 6.39e-02 | 5.32e-06 | 8 |
| 28 | 2.32e-01 | 1.08e-01 | 3.35e-02 | 3.12e-01 | 2.37e-01 | 3.47e-02 | 1.52e-01 | 7.46e-02 | 8.12e-06 | 8 |
| 29 | 1.90e-02 | 1.84e-01 | 1.02e-02 | 2.49e-01 | 7.19e-02 | 2.00e-02 | 3.17e-01 | 2.50e-06 | 5.57e-02 | 7 |
| 30 | 2.07e-02 | 2.03e-01 | 1.15e-02 | 2.41e-01 | 6.79e-02 | 2.11e-02 | 3.68e-01 | 2.91e-06 | 6.44e-02 | 7 |
| 31 | 2.04e-02 | 2.02e-01 | 1.14e-02 | 2.40e-01 | 6.74e-02 | 2.10e-02 | 3.66e-01 | 3.05e-06 | 6.41e-02 | 7 |
| 32 | 7.07e-02 | 2.55e-01 | 2.04e-02 | 3.09e-01 | 1.20e-01 | 2.74e-02 | 4.17e-01 | 8.74e-06 | 7.41e-02 | 7 |
| 33 | 5.48e-02 | 1.56e-01 | 1.22e-06 | 2.80e-01 | 1.11e-01 | 1.89e-02 | 2.44e-01 | 1.00e-02 | 2.53e-02 | 2 |
| 34 | 6.04e-02 | 1.65e-01 | 3.09e-06 | 2.67e-01 | 1.07e-01 | 1.76e-02 | 2.79e-01 | 1.14e-02 | 2.83e-02 | 2 |
| 35 | 5.99e-02 | 1.64e-01 | 3.10e-06 | 2.66e-01 | 1.06e-01 | 1.75e-02 | 2.78e-01 | 1.14e-02 | 2.82e-02 | 2 |
| 36 | 1.61e-01 | 2.32e-01 | 1.20e-05 | 3.85e-01 | 2.14e-01 | 3.90e-02 | 3.26e-01 | 2.06e-02 | 3.35e-02 | 2 |

Table 10: Training MSE for Trajectories by each Neural Network and Label Assignment

# G  Environment Assignment Convergence on GS

Figure 6 presents the temporal evolution of environment assignment probabilities using the LEADS base model on GS.

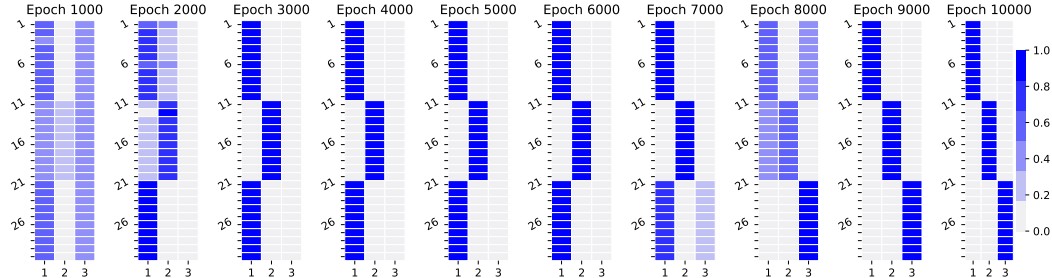

Figure 6: Environment Assignment Probability over Time with LEADS as Base Model on GS. Despite initial inaccuracies due to complex dynamics, the assignment ultimately converges to the correct label.

# H  Plots on Learned Dynamics

We present the recovered test trajectories produced by the learned neural network. Figures 7, 8, and 9 illustrate GS; Figures 10, 11, and 12 depict NS; and Figure 13 shows LV. A close examination reveals that the trajectories predicted by DynaInfer closely align with both the Oracle and the ground truth for the selected systems.

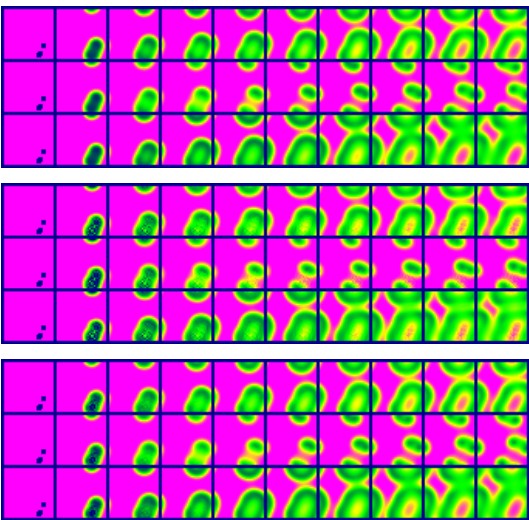

Figure 7: Comparison of final GS states predicted by DynaInfer (bottom) and Oracle (middle) against the ground truth (top) with base model LEADS.

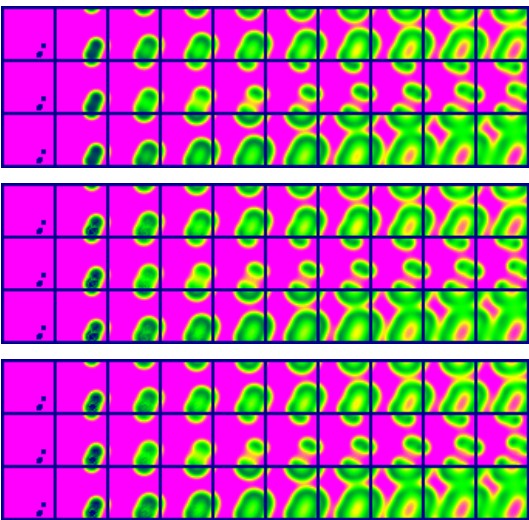

Figure 8: Comparison of final GS states predicted by DynaInfer (bottom) and Oracle (middle) against the ground truth (top) with base model CoDA-$l_1$.

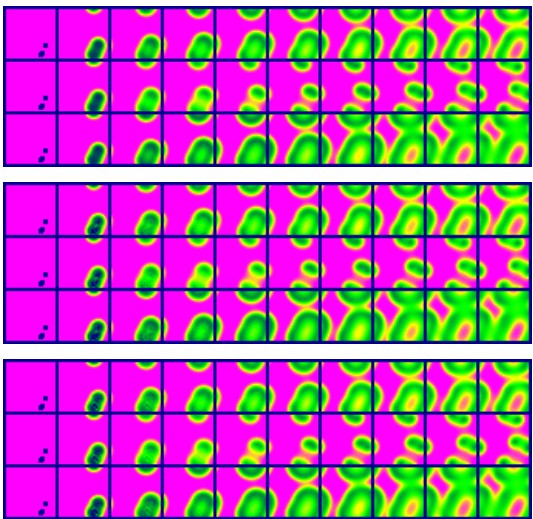

Figure 9: Comparison of final GS states predicted by DynaInfer (bottom) and Oracle (middle) against the ground truth (top) with base model CoDA-$l_2$.

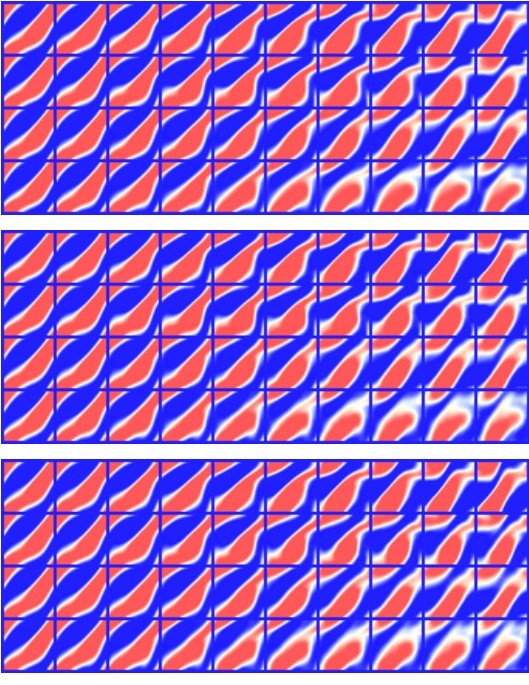

Figure 10: Comparison of final NS states predicted by DynaInfer (bottom) and Oracle (middle) against the ground truth (top) with base model LEADS.

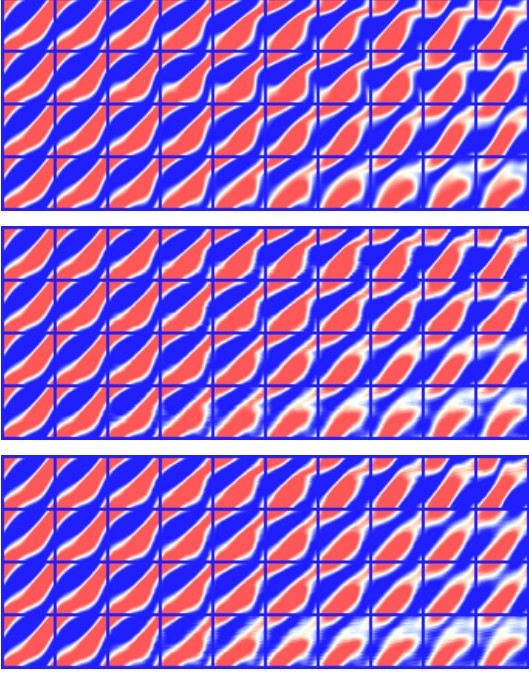

Figure 11: Comparison of final NS states predicted by DynaInfer (bottom) and Oracle (middle) against the ground truth (top) with base model CoDA-$l_1$.

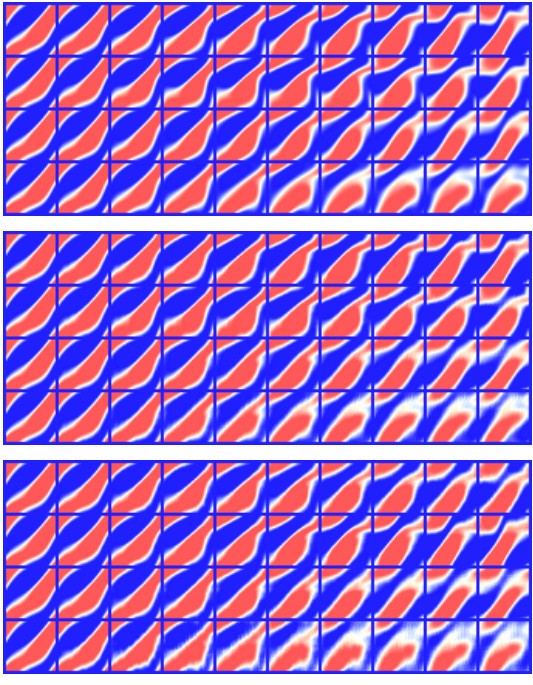

Figure 12: Comparison of final NS states predicted by DynaInfer (bottom) and Oracle (middle) against the ground truth (top) with base model CoDA-$l_2$.

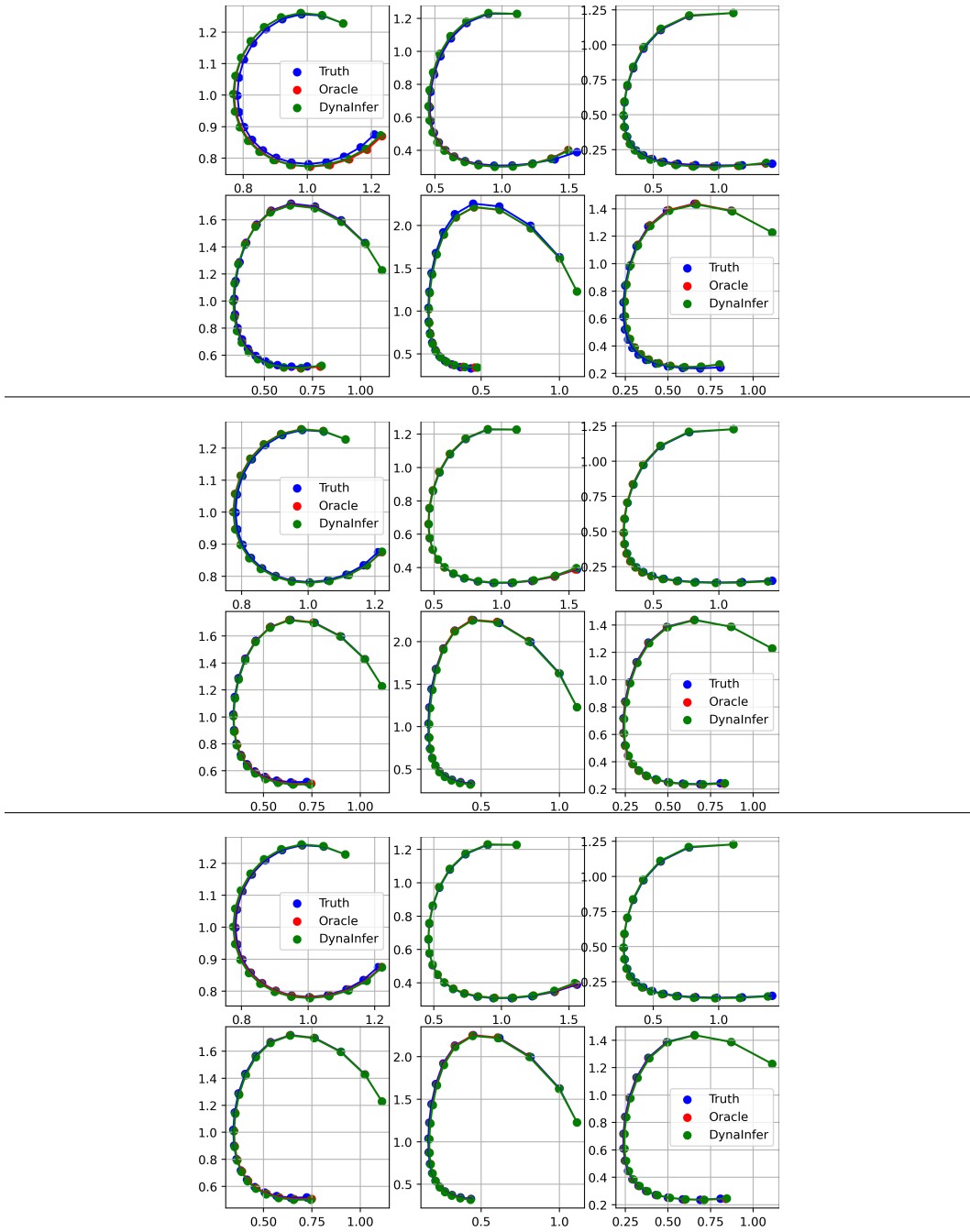

Figure 13: Comparison of predicted LV trajectories from 6 environments by DynaInfer (green) and Oracle (red) against the ground truth (blue) with base models LEADS (top), CoDA-$l_1$ (middle), and CoDA-$l_2$ (bottom).

