# OpenReview forum: "Environment Inference for Learning Generalizable Dynamical System"
_NeurIPS.cc/2025/Conference — NeurIPS 2025 spotlight_

### Official Review · Reviewer_sBk5 · 2025-06-16

**Clarity:** 1
**Significance:** 2
**Originality:** 3
**Rating:** 4
**Confidence:** 3

**Summary:**

The authors introduce DynaInfer, a method to infer environment variables for dynamical systems directly from training data. This method allows to use multi-environment data and predict their dynamic response more accurately by inferring the environment variable. Experiments show improvments over other basic environment inference strategies.

**Questions:**

Apart from the questions within the section above, there are the following comments:
- order of presentation:
  - Algorithm 1: this is presented before equation (4) and (5), so it is hard to understand without having seen these equations.
  - line 167: Proposition 3.1 mentioned two subsections before it is introduced
- Proposition 3.1: is there a restriction on $C\leq 0$? Otherwise your propostion does not provide any useful information
- line 208: "at test time, ...". This is unclear. Please explain.
- line 210" Do you fine tune on data where you test or do you split the test data for fine tuning and for evaluation?
- Line 225: What is LEADS, CoDA-l1/2? How can they be combined with your method?
- Line 243: how can your method outperform the oracle?
- Table 1: is the first colum in LEADS / CoDA-l1/2 the Train MSE or what metric is displayed there?

Smaller comments:
- line 85: ERM abbreviation not introduced
- line 99: unclear what is $g$
- Table 1 and 2: font size is too small. Keep same as in text preferrably.
- Fig 3 is never referenced but fig 5 instead

**Ethical Concerns:**

["NO or VERY MINOR ethics concerns only"]

**Final Justification:**

The authors have addressed all my concerns. I do hope that my main concern about the presentation will be sufficiently adressed, which cannot be shown to all extend within the rebuttal phase.

**Limitations:**

yes

**Quality:**

2

**Strengths And Weaknesses:**

Strengths:
- The method is simple and easy to follow with great foundations in clustering algorithms
- The experiments show promising results
- The discussion is insightful

Weaknesses:
- Presentation:
  - In the abstract and introduction the problem formulation is unclear. You never mention the multi-environment setting. A motivating example with a clear figure would be of great value.
  - Similarly, the environment variable is unclear. Is this some abstract latent variable or does it has physical meaning?
  - The significance and importance of the problem should be worked out more clearly. Which practical settings does your method solve now.
- Method:
  - in line 155/156 you discuss that "neural networks [...] effectively operates analogously to the "centroid" ...". It would be important to show this observation in some form of experiment to have a strong basis for your method.
  - For convergence speed your method seems to rely on a good initialisation. Please clarify its effect and which scheme you choose.
- Experiments:
  - It is unclear what LEADS and CoDA-l1/2 are doing and how it can be combined with your method.
  - There are not baseline methods except for the basic environment variable assignments that your present. In the related work you present some baseline methods. A rigorous comparison would be necessary within the experiments

---

> ### Author Rebuttal · Authors · 2025-07-29
>
> We sincerely appreciate the reviewer’s time and insightful feedback on our work. Below, we address the raised concerns point by point.
>
> >  **[W1-a] In the abstract and introduction the problem formulation is unclear. You never mention the multi-environment setting. A motivating example with a clear figure would be of great value**
>
> **Response:**
> - We thank the reviewer for this valuable suggestion. We have now **clarified the multi-environment setting** in both the abstract and introduction, as in Lines 29-33:
>   - "Recent work in dynamical systems addresses this by introducing a multi-environment setting, where trajectories follow distinct dynamics across environments. These studies developed generalization methods that learn a shared global component while accounting for environment-specific variations, avoiding the limitations of underperforming averaged models."
> - We have **added a new figure** to provide a motivating example in the revised manuscript.
>
> >  **[W1-b] Similarly, the environment variable is unclear. Is this some abstract latent variable or does it has physical meaning?**
>
> **Response:**
> -  The environment variable (or environment label) in our work is an abstract latent variable that indicates which environment a given trajectory belongs to.
> -  Each environment has distinct underlying system dynamics, and both the dynamics and environment labels are unknown a priori—our method learns them jointly.
> - We **have clarified this** in the revised paper in Lines 36-37:
>   - "These environment labels enable algorithms to identify and exploit both similarities and differences across environments."
>
> >  **[W1-c] The significance and importance of the problem should be worked out more clearly. Which practical settings does your method solve now.**
>
> **Response:**
> We sincerely thank the reviewer for raising this important point.
>
> - **Theoretical & Practical Significance:** Our work addresses a fundamental challenge in machine learning—OOD generalization for dynamic systems—with broad theoretical and real-world implications. Unlike standard OOD settings that assume known environment labels, we focus on the more realistic yet understudied scenario where such labels are unavailable.
>
> - **Key Motivations & Real-World Relevance:** The absence of environment labels in dynamic systems often arises due to inherent complexities, including:
>   - **Data Collection Constraints:** In ecological studies, critical environmental factors (e.g., location and height) could be unrecorded.
>   - **Multi-Source Data Aggregation:** In distributed sensor networks, data from heterogeneous sources often lack consistent environment annotations.
>   - **Privacy & Regulatory Barriers:** In healthcare or finance, sharing environment-specific metadata (e.g., patient demographics) is frequently restricted.
>
> By formalizing and solving this unlabeled OOD setting, our method bridges a critical gap in deploying robust dynamic systems in real-world applications. We **have expanded these points** in the revised manuscript to better emphasize both the novelty and applicability of our approach.
>
> >  **[W2-a]  in line 155/156 you discuss that "neural networks [...] effectively operates analogously to the "centroid" ...". It would be important to show this observation in some form of experiment to have a strong basis for your method.**
>
> **Response:**
> We thank the reviewer for this insightful suggestion. To empirically support our observation, we conducted an additional experiment measuring the MSE loss for each training sample across different neural networks (NNs). The results (samples shown below for brevity and **full table now included in the Appendix**) reveal two key findings:
> 1. **Each training sample has a uniquely best-fit NN** (i.e., one NN achieves minimal loss for that sample).
> 2. **This NN also generalizes to a cluster of trajectories**—specifically, those originating from the same environment.
>
> |Traj.|NN1|NN2|NN3|NN4|NN5|NN6|NN7|NN8|NN9|Assign.|
> |---|---|---|---|---|---|---|---|---|---|---|
> |1|2.1E-1|1.2E-1|2.8E-1|7.8E-6|8.3E-2|1.6E-1|3.1E-1|2.5E-1|2.4E-1|3|
> |2|2.1E-1|1.2E-1|2.7E-1|9.4E-6|8.4E-2|1.5E-1|3.4E-1|2.4E-1|2.3E-1|3|
> |3|2.1E-1|1.2E-1|2.6E-1|1.0E-5|8.4E-2|1.5E-1|3.3E-1|2.4E-1|2.3E-1|3|
> |4|2.5E-1|1.5E-1|3.8E-1|2.0E-5|1.1E-1|1.9E-1|4.0E-1|3.1E-1|3.1E-1|3|
> |5|3.3E-2|1.6E-1|1.1E-1|8.3E-2|5.4E-6|5.4E-2|3.7E-1|7.3E-2|1.4E-1|4|
> |6|3.2E-2|1.9E-1|1.1E-1|8.5E-2|4.3E-6|5.6E-2|4.2E-1|6.8E-2|1.4E-1|4|
> |7|3.1E-2|1.9E-1|1.1E-1|8.5E-2|4.5E-6|5.5E-2|4.2E-1|6.7E-2|1.4E-1|4|
> |8|2.8E-2|2.7E-1|2.1E-1|1.1E-1|2.2E-5|9.6E-2|5.4E-1|1.2E-1|2.4E-1|4|
> |...|...|...|...|...|...|...|...|...|...|...|
> |29|1.9E-2|1.8E-1|1.0E-2|2.5E-1|7.2E-2|2.0E-2|3.2E-1|2.5E-6|5.6E-2|7|
> |30|2.1E-2|2.0E-1|1.1E-2|2.4E-1|6.8E-2|2.1E-2|3.7E-1|2.9E-6|6.4E-2|7|
> |31|2.0E-2|2.0E-1|1.1E-2|2.4E-1|6.7E-2|2.1E-2|3.7E-1|3.0E-6|6.4E-2|7|
> |32|7.1E-2|2.6E-1|2.0E-2|3.1E-1|1.2E-1|2.7E-2|4.2E-1|8.7E-6|7.4E-2|7|
> |33|5.5E-2|1.6E-1|1.2E-6|2.8E-1|1.1E-1|1.9E-2|2.4E-1|1.0E-2|2.5E-2|2|
> |34|6.0E-2|1.7E-1|3.1E-6|2.7E-1|1.1E-1|1.8E-2|2.8E-1|1.1E-2|2.8E-2|2|
> |35|6.0E-2|1.6E-1|3.1E-6|2.7E-1|1.1E-1|1.7E-2|2.8E-1|1.1E-2|2.8E-2|2|
> |36|1.6E-1|2.3E-1|1.2E-5|3.8E-1|2.1E-1|3.9E-2|3.3E-1|2.1E-2|3.4E-2|2|
>
> This alignment between "centroid-like" NN behavior and environment-specific trajectory clusters reinforces our methodological rationale. We have **added this analysis and extended our discussion** in the revised manuscript.
>
> >  **[W2-b] For convergence speed your method seems to rely on a good initialisation. Please clarify its effect and which scheme you choose.**
>
> **Response:**
> - All NN params in our method are actually **randomly initialized**. The convergence speed differences observed in **Figure 3** (e.g., LV learning faster than GS) arise from inherent task complexity rather than initialization.
> - LV’s simpler dynamics lead to rapid convergence (e.g., stable assignments before epoch 50), even when initial assignments are random.
> - We **have clarified this point in the revision** to avoid potential misunderstandings.
>
> >  **[W3-a] & [Q5] It is unclear what LEADS and CoDA-l1/2 are doing and how it can be combined with your method.**
>
> **Response:**
> Our method addresses **environment inference** for dynamic system generalization, which requires building upon an existing OOD generalization model for dynamic systems. We adopt LEADS and CoDA-l1/2 as our base OOD generalization models because they are well-established in this field.  We **have clarified this integration** in the revised manuscript.
>
>
> >  **[W3-b] There are not baseline methods except for the basic environment variable assignments that your present. In the related work you present some baseline methods. A rigorous comparison would be necessary within the experiments.**
>
> **Response:**
> - Our work is the **first** to address **environment inference** for unlabeled trajectory data in dynamic system generalization. Since this problem has not been studied before, no direct assignment baselines exist for comparison.
> - While we discuss related methods for generalization in the related work, these are not designed for environment inference—a key distinction highlighted in our paper.
> - For clarity, we have **explicitly emphasized our contribution** in the revised manuscript, distinguishing our problem setting from prior work.
>
> >  **[Q1] Order of presentation: 1) Algorithm 1 [...] & 2) Proposition 3.1**
>
> **Response:**
> We appreciate the feedback. We **have reordered the presentation**.
>
> >  **[Q2] Proposition 3.1: is there a restriction on $c\leq0$?**
>
> **Response:**
> Thank you for catching this. The proposition requires $C > 0$ to ensure the loss decreases by a positive constant. We’ve corrected this in the revision.
>
> >  **[Q3] line 208: "at test time, ...". This is unclear. Please explain.**
>
> **Response:**
> Thank you for pointing this out. Instead of assuming test environment labels are known, we are dealing with a greater challenge that test label is also unknown. In this case, the label is inferred by comparing predicted vs. test trajectories over the first temporal interval. We **have now clarified** in the revised paper.
>
> >  **[Q4] line 210" Do you fine tune on data where you test or do you split the test data for fine tuning and for evaluation?**
>
> **Response:**
> Our method follows standard domain adaptation practice that finetunes on the test domain data. We **have clarify this point** in the revision.
>
> >  **[Q6] Line 243: how can your method outperform the oracle?**
>
> **Response:**
> Thank you for pointing this out,
> Our method can outperform the oracle due to three key factors:
> - Our optimized environment labels may better capture generalization patterns than predefined oracle labels
> - Real-world trajectory overlaps make rigid oracle assignments suboptimal. Certain trajectories may lie closer to clusters in other environments than their assigned oracle labels suggest.
> - By optimizing environment assignments during training, our method resolves label ambiguities adaptively, prioritizing generalizable dynamics over strict adherence to oracle label fidelity.
> We **have now clarify this effect** in the revision.
>
> >  **[Q7] Table 1: is the first colum in LEADS / CoDA-l1/2 the Train MSE or what metric is displayed there?**
>
> **Response:**
> The first column shows the training MSE. We **have clarified this** in the revised table header.
>
> >  **[Smaller comments] 1) line 85: ERM abbreviation; 2) line 99: unclear what is g; 3) Table 1 and 2: font size is too small. 4) Fig 3 is never referenced but fig 5 instead**
>
> **Response:**
> 1. ERM is the expected risk minimization. We **have now defined** the ERM in the revision.
> 2. $g$ stands for the neural network learning the individual part. we **have now clarified** $g$ in the revision.
> 3. We **have adjusted the font size** in table 1 and 2.
> 4. We **have fixed this latex reference mistake**.
>
> Again, we sincerely appreciate your valuable comments and welcome further discussion.

---

> > ### Comment · Reviewer_sBk5 · 2025-08-03
> >
> > I thank the authors for their detailed answers to my concerns and questions. I believe you have addressed all my points sufficiently and trust that my main concern about the presentation of the paper including its problem formulation and motivation will be addressed as promised. I have adapted my score accordingly.

---

> > > ### Author Response · Authors · 2025-08-03
> > >
> > > We sincerely appreciate your thoughtful feedback and kind evaluation of our work. Your suggestion is invaluable in helping us improve, and it will certainly guide our future research. If you have any further insights, we would be very happy to hear. Thank you! :)

---

### Official Review · Reviewer_dkAK · 2025-06-29

**Clarity:** 2
**Significance:** 3
**Originality:** 3
**Rating:** 5
**Confidence:** 3

**Summary:**

This work presents DynaInfer, a bi-level algorithm for learning environment labels and generalized environmental dynamics at the same time. This work is motivated by the fact that the collected trajectories generated from various environments lack true labels, and thus, it is hardly possible to use them for training generalizable surrogate models modulated on such labels. Instead of using such ground truth labels, this work suggests to optimize such labels altogether during optimization. This framework is bi-level, because it optimizes the label assignments to the trajectories in the first stage, and then optimizes the surrogate model parameters in the second stage based on the label assignments. This process is analogous to the K-means clustering, as the authors explain. The authors provide theoretical proof about the monotonic decrease of the approximation error using this bi-level approach. Then, the authors provide experimental results on three dynamical systems, which show DynaInfer’s efficacy over the baseline label assignment approaches.

**Questions:**

In my understanding, the number “M” is a hyperparameter that interpolates between “All-in-one” (M=1) and “One-per-Env” (M=# trajectory) strategy. If it is, I think it would be also interesting to see how the performance interpolates between them based on M. Then, I think it becomes easier to claim that DynaInfer is superior than both of these strategies as they are lower bounds of DynaInfer’s performance.

**Ethical Concerns:**

["NO or VERY MINOR ethics concerns only"]

**Final Justification:**

This paper solves the problem when there is no ground truth labels for the trajectories, which seems to be a fairly new problem in this field. Considering such contribution and as the authors provided detailed responses to my concerns, I will raise my score to accept.

**Limitations:**

As I pointed out, the computational cost and the presence of hyperparameter M seems to be a potential limitations of this work.

**Paper Formatting Concerns:**

No major formatting issues.

**Quality:**

3

**Strengths And Weaknesses:**

Strengths

- The authors provided a theoretical analysis of their algorithm, and the experimental results look good. DynaInfer consistently outperforms the other baseline label assignment strategies, and the figure about convergence looks neat.

Weaknesses

- I feel some clarifications are needed about methodology. Specifically, in section 3.1, how do we solve the minimization problem posed in Eq. 4? Do we just try every label and select the best one? If that is the case, I believe there should be a computational cost analysis, because such brute force approach would incur more overhead. I think then the computational cost would increase linearly to M, which could be problematic if we use DynaInfer for a large dataset.
- Even though the authors mentioned that DynaInfer consistently outperforms the other baseline label assignment strategies even when they use underestimated “M”, it seems not to be true. For instance, when M = 3, LEADS performance for LV is 2.47E-2, which is worse than the “One-per-Env” strategy (4.15E-4). I’m a little bit concerned about this necessity for prior knowledge of “M”. To truly make this approach free from prior knowledge, it would be good if there is one more strategy to optimize “M” automatically (even though the authors suggested a manual strategy for it).
- It is not a main weakness, but it would be better to highlight the difference between the results in Figure 2 to show how DynaInfer outperforms the other methods.

---

> ### Author Rebuttal · Authors · 2025-07-29
>
> We sincerely appreciate the reviewer’s time and insightful feedback on our work. Below, we address the raised questions and concerns point by point. For clarity, some responses have been condensed to highlight key points. We welcome further discussion and appreciate any additional suggestions.
>
> >  **[W1] I feel some clarifications are needed about methodology. Specifically, in section 3.1, how do we solve the minimization problem posed in Eq. 4? Do we just try every label and select the best one? If that is the case, I believe there should be a computational cost analysis, because such brute force approach would incur more overhead. I think then the computational cost would increase linearly to M, which could be problematic if we use DynaInfer for a large dataset.**
>
> **Response:**
> We thank the reviewer for raising this crucial implementation concern. Below we clarify DynaInfer’s computational efficiency and justify its design trade-offs:
>
> #### 1. **Optimization Strategy**
> - While the reviewer correctly notes that inferring environment labels involves evaluating trajectories across multiple networks, this process is fully parallelized across both environments and batches, avoiding sequential overhead.
> - DynaInfer’s alternating optimization comprises two stages:
>   - Inference Stage: Complexity scales with assumed number of environments (M), and batch size (N), but crucially, it avoids the far greater cost of computing the full training graph (as required in optimization stage).
>   - Optimization Stage: Complexity scales linearly with (N).
>
> #### 2. **Empirical Runtime Analysis**
> We have included a wall-clock time comparison (seconds) on the LV system using LEADS as the base model, evaluating DynaInfer against the Oracle baseline (which requires no inference). The DynaInfer results are shown for the standard setting (*M*=9) and an exaggerated scenario (*M*=36) to stress-test efficiency:
>
> | Setting       | Oracle | M=9   | M=36  |
> |--------------|--------|-------|-------|
> | Time (s)     | 2526| 3258 | 4849 |
> | Overhead     | —      | +28.9% vs. Oracle | +48.9% vs. M=9 |
>
> - **Justifiable Overhead**: DynaInfer incurs only a **28.9% runtime overhead** (3258s vs. Oracle’s 2526s) under the real-world *M*=9 setting. This modest cost is outweighed by DynaInfer’s ability to **match Oracle performance *without* labeled data** (Tables 1–2, main text), eliminating manual labeling entirely.
> - **Sub-linear Scaling**: Even under the exaggerated *M*=36 scenario (4× the true environment count, and not practical in real world), runtime increases by only **48.9%** versus *M*=9.
>
> #### 3. **Future Strategies**
> - We agree with the reviewer that runtime is worth optimizing. In the revised manuscript, we have expanded discussion on efficiency:
>    - *Adaptive Early Stopping*: Halting inference once label assignments stabilize.
>    - *Dynamic Batching*: Prioritizing uncertain trajectories for re-evaluation.
> - These directions could reduce overhead while retaining DynaInfer’s benefits. We thank the reviewer for highlighting this nuance and will ensure above discussions are explicitly included in the final draft.
>
> >  **[W2] Even though the authors mentioned that DynaInfer consistently outperforms the other baseline label assignment strategies even when they use underestimated “M”, it seems not to be true. [...]. I’m a little bit concerned about this necessity for prior knowledge of “M”. To truly make this approach free from prior knowledge, it would be good if there is one more strategy to optimize “M” automatically (even though the authors suggested a manual strategy for it).**
>
> **Response:**
> We sincerely appreciate the reviewer’s insightful critique regarding the need for prior knowledge of M. We address these concerns below and propose concrete improvements:
>
> #### 1. **Correction and Clarification**
> -  We acknowledge the reviewer’s observation regarding the performance of DynaInfer with underestimated *M*. Our original claim inadvertently omitted the edge case of the *One-per-Env* baseline, which requires *M* to match the number of trajectories—a scenario that incurs prohibitive computational costs and contradicts the goal of domain generalization in dynamic systems. We apologize for this oversight.
> - The corrected statement in the manuscript now reads: “Lastly, DynaInfer consistently outperforms other non-oracle approaches when M is underestimated, except for the One-per-Env baseline (which requires M equal to the number of trajectories and is computationally infeasible).”
>
> #### 2. **Possible Automatic M Identification Strategy**
> To reduce reliance on prior knowledge of *M*, we’ve identified a practical and automatic strategy:
> - **Gradual Increment with Early Validation:** We observe that an appropriate M yields measurable improvements even in early training stages (e.g., within the first 500 steps). Thus, M can be incrementally increased while monitoring validation performance, stopping when improvements plateau. This approach balances efficiency with robustness, avoiding the need for exhaustive manual tuning.
>
> #### 3. **Empirical Support for Robustness over M**
> We conducted additional experiments on the LV system to evaluate robustness to *M* by setting *M* from 5 to 36 (which is the number of trajectories).
>
> | M        | 5    | 6    | 7  | 8    | 9    | 10  | 11 | 12   | 13   | 14   | 15 | 16   | 17   | 18  | 19 | 20   |
> |----------|------|------|----|------|------|-----|----|------| ------|------|----|------|------|-----|----|------|
> | **# Labels** | 5    | 6    | 7  | 8    | 9    | 8.8 | 9  | 10.2 | 9    | 9    | 10 | 10.8 | 10   | 11  | 11 | 10   |
>
> | M        | 21   | 22   | 23 | 24   | 25   | 26  | 27 | 28   | 29   | 30   | 31 | 32   | 33   | 34  | 35 | 36   |
> |----------|------|------|----|------|------|-----|----|------|------|------|----|------|------|-----|----|------|
> | **# Labels** | 10.8 | 11   | 9  | 10   | 10.2 | 11  | 10 | 10   | 10   | 11.2 | 10 | 9.2  | 10   | 10.8  | 10 | 11   |
>
> - As shown below (results averaged over 5 runs), DynaInfer adapts to the overestimation of *M*: The learned label count converges to a reasonable range for *M* (8–11), closely aligning with the true condition (true *M* = 9 in the LV system).
> - This also support the viability of automatic M selection.
>
> We **have added such table and discussion** in revision.
>
> >  **[W3] It is not a main weakness, but it would be better to highlight the difference between the results in Figure 2 to show how DynaInfer outperforms the other methods.**
>
> **Response:**
> We sincerely appreciate the reviewer’s constructive suggestion to more clearly demonstrate DynaInfer’s advantages over baseline methods.
>
> - In our original submission, Figure 2 primarily focused on comparing DynaInfer’s performance against the *Oracle* baseline to emphasize its near-optimal capability, while omitting other methods for visual clarity (as their results were substantially weaker).
> - However, we fully agree that explicitly quantifying this performance gap would strengthen the paper’s impact.  Accordingly, we **have included a detailed visualization** in the revised Appendix to clearly showcase the differences.
>
> >  **[Q1] In my understanding, the number “M” is a hyperparameter that interpolates between “All-in-one” (M=1) and “One-per-Env” (M=# trajectory) strategy. If it is, I think it would be also interesting to see how the performance interpolates between them based on M. Then, I think it becomes easier to claim that DynaInfer is superior than both of these strategies as they are lower bounds of DynaInfer’s performance.**
>
> **Response:**
> We sincerely appreciate the reviewer's insightful suggestion regarding the interpolation behavior of M. We have conducted extensive experiments across the full spectrum of M values (from 1 to the number of trajectories) on the LV system, with results that strongly support our claims.:
>
> |M|1|2|3|4|5|6|7|8|9|
> |---|---|---|---|---|---|---|---|---|---|
> |MSE|7.40E-2|4.01E-2|2.48E-2|1.04E-2|7.13E-3|4.24E-3|2.63E-3|8.25E-4|5.99E-5|
>
> |M|10|11|12|13|14|15|16|17|18|
> |---|---|---|---|---|---|---|---|---|---|
> |MSE|2.45E-4|2.33E-4|2.07E-4|1.63E-4|2.24E-4|1.15E-4|1.76E-4|1.83E-4|3.60E-4|
>
> |M|19|20|21|22|23|24|25|26|27|
> |---|---|---|---|---|---|---|---|---|---|
> |MSE|2.60E-4|1.36E-4|1.21E-4|2.45E-4|9.06E-5|2.38E-4|8.27E-5|1.06E-4|3.71E-4|
>
> |M|28|29|30|31|32|33|34|35|36|
> |---|---|---|---|---|---|---|---|---|---|
> |MSE|2.04E-4|3.65E-4|1.42E-3|3.84E-4|1.16E-4|3.89E-4|3.10E-4|1.72E-4|2.77E-4|
>
> - As shown in the tables below, MSE decreases sharply from M=1 (All-in-One) to M=9 (true number of environments), then stabilizes. The optimal performance occurs at M=9 and performance remains robust even with M overshoot.
> - The performance on the LV for different M, interpolates between M=1 and M=#trajectory. Similar trends were observed in the GS dataset (to be included in the appendix).
> - We would claim that DynaInfer is superior than both of these strategies as they are lower bounds of DynaInfer’s performance. We will add a new figure showing the performance-M curve for all datasets in the revised manuscript.
>
> Finally, we sincerely welcome further discussion and appreciate any additional suggestions.

---

> > ### Comment · Reviewer_dkAK · 2025-08-03
> >
> > I appreciate the authors' detailed responses and additional experimental results. I believe these additional information would be valuable in improving the paper quality. I will raise my score to accept, as these responses resolved most of my concerns.

---

> > > ### Author Response · Authors · 2025-08-03
> > >
> > > We sincerely appreciate your insightful feedback and positive assessment of our work. Your input is invaluable in helping us enhance our current efforts and guide future works. If you have any further suggestions, please don’t hesitate to reach out. Thank you! :)

---

### Official Review · Reviewer_ghmr · 2025-07-02

**Clarity:** 3
**Significance:** 2
**Originality:** 2
**Rating:** 5
**Confidence:** 2

**Summary:**

The paper studies the problem of learning dynamical system models wherein the model is adapted to data coming from different environments. This is parameterized as a leraning problem wherein there are common parameters that are environment-independent and some environment dependent parameters. However, the environment labels for trajectories are latent and need to be inferred. The paper uses a k-means (or also EM-style) approach to solve this problem by alternating between learning an environment assignment and at the same time using the current environment assignment to learn new parameters. This is very much in line with k-means clustering. Some ideas from clustering are used to improve the learning scheme. It is demonstrated on a series of interesting benchmark examples.

**Questions:**

I would welcome your observations on some of the issues raised in my review.

**Ethical Concerns:**

["NO or VERY MINOR ethics concerns only"]

**Final Justification:**

Given the discussions surrounding the paper, I raised my score to accept but also reduced the confidence of my review. Clearly, I did not understand some of the original aspects of this paper.

**Limitations:**

I did not see a meaningful discussion but the paper is written in a careful and modest style that makes limitations clear as one reads.

**Quality:**

2

**Strengths And Weaknesses:**

The ideas in the paper are very interesting: I really enjoyed reading about the problem formulation and its translation into an alternating minimization framework. I found the writing to be crystal clear and the ideas to be very easy to comprehend.  The results are also very interesting.

Perhaps, my biggest critique is that the paper ignores a lot of research into identical ideas from the world of switched systems regression.  Perhaps the connection is not well known, but the problem of learning switched systems from data is very similar if not identical in many respects. Lauer and his coworkers have been working on a very similar idea that they call k-linear regression -- it is principally meant for linear systems or learning non linear systems once the basis functions are identified. However, the ideas are very similar. I would recommend looking into Lauer's papers. These ideas are somewhat mainstream in the area of switched systems identification.

@article{lauer:hal-00743954,
  TITLE = {{Estimating the probability of success of a simple algorithm for switched linear regression}},
  AUTHOR = {Lauer, Fabien},
  URL = {https://hal.science/hal-00743954},
  JOURNAL = {{Nonlinear Analysis: Hybrid Systems}},
  PUBLISHER = {{Elsevier}},
  VOLUME = {8},
  PAGES = {31-47},
  YEAR = {2013},
  DOI = {10.1016/j.nahs.2012.10.001},
  KEYWORDS = {switched regression ; system identification ; switched linear systems ; piecewise affine systems ; sample size ; nonconvex optimization ; global optimality},
  PDF = {https://hal.science/hal-00743954v1/file/LauerNAHS12.pdf},
}

You can find more details in the Springer book of Lauer and Bloch:

Hybrid System Identification: Theory and Algorithms for Learning Switching Models -- Lauer and Bloch, 2019.

@article{BIANCHI2022110589,
title = {A constrained clustering approach to bounded-error identification of switched and piecewise affine systems},
journal = {Automatica},
volume = {146},
pages = {110589},
year = {2022},
issn = {0005-1098},
doi = {https://doi.org/10.1016/j.automatica.2022.110589},
}


Although I agree that the setting of this paper is different from that of switched system (the label assignment is stationary in time in this paper whereas in switched system regression, it varies over time), I am just interested in understanding how the overall ideas connect -- the connections between the two settings are stronger than the differences.

- Also, the optimization approach used here is well known to be problematic. A lot of studies have shown that for k-means regression. But specifically this sort of alternating minimization has issues. It can be shown to seldom converge to a KKT point and often gets stuck at saddle points. See this paper by Helton and Merino for an example of such a result --
 Helton, J., Merino, O.: Coordinate optimization for bi-convex matrix inequalities. In: IEEE Conf. on Decision & Control(CDC). p. 36093613 (1997)

---

> ### Author Rebuttal · Authors · 2025-07-30
>
> We sincerely appreciate the reviewer’s time and valuable feedback on our work. Below, we provide a point-by-point response to their comments. For clarity, some raised questions have been condensed to focus on key points.
>
> >  **[W1] Perhaps, my biggest critique is that the paper ignores a lot of research into identical ideas from the world of switched systems regression. Perhaps the connection is not well known, but the problem of learning switched systems from data is very similar if not identical in many respects. Lauer and his coworkers have been working on a very similar idea that they call k-linear regression -- it is principally meant for linear systems or learning non linear systems once the basis functions are identified. However, the ideas are very similar. I would recommend looking into Lauer's papers. These ideas are somewhat mainstream in the area of switched systems identification.**
>
> **Response:**
> We sincerely appreciate the reviewer’s insightful feedback and the valuable references to the switched systems literature, particularly the work by Lauer. We have carefully reviewed these works and provide below a detailed discussion of their connections to and distinctions from our approach.
>
> #### **1. Key Distinctions Between Problem Settings**
> - **Switched System Learning (SSL)** primarily addresses:
>   - Identification of systems with discrete transitions between predefined subsystems.
>   - The switching mechanism is either known or learned as part of the modeling.
>   - Modeling of relatively simple subsystem dynamics (typically linear or affine).
>   - The primary objective is jointly estimating subsystem parameters and switching logic.
>
> - **Our Dynamic System Learning Generalization Work** differs fundamentally in that:
>   - **Absence of Switching** Our framework learns vector field functions (ODE or PDE) for continuous dynamical systems without switching; while SSL does;
>   - **Out-of-distribution and In-distribution**: Our primary focus is out-of-distribution generalization rather than in-distribution fitting, while SSL does;
>        -  Therefore, our optimization objective incorporate additional regularization terms specifically designed for domain generalization.
>   - **Non-linear Dynamics**: Our approach handles arbitrary nonlinear dynamics with complex boundary conditions (e.g., Navier-Stokes systems as in our work), while SSL typically handles switched linear or affine systems;
>
> #### **2. Technical Comparisons**
> While both lines of approaches employ clustering-like mechanisms, there are crucial differences:
>
> - **Different Clustering**:
>      -  Switched systems methods (e.g., Lauer, Bianchi) cluster individual data points to identify subsystem boundaries.
>      - Our method clusters entire trajectories to learn the shared part that governs generalizable dynamics.
> - **Different Objectives**:
>      - The optimization objectives are fundamentally different (fitting accuracy vs.  domain generalization capability)
>
> #### **3. Potential Connections and Future Directions**
>
> We acknowledge several insightful connections in the suggested papers:
> - The *k*-LinReg in Lauer's work also employs a min-min scheme, transforming the original least-squares switched linear regression problem into a minimum-of-error formulation. While our objective differs (leading to distinct theoretical guarantees), the overall methodology is analogous. Additionally, we could explore probabilistic evaluations of success rates for global optimality, inspired by their work, as future research.
> - The two-phase constrained clustering approach by Bianchi introduces preference-space clustering for switched systems, which could inspire improved initialization for our environment discovery scheme. However, since their method computes preference vectors over individual data points, direct application to our setting—where we handle trajectories governed by ODEs/PDEs—is infeasible. Nevertheless, adapting their approach by considering the likelihood in temporal differences among trajectories could be promising, and we will explore this direction in future work.
>
> #### **4. Additions to Manuscript**
> Considering the insightful connections in switched systems, we would:
> - Discuss these connections and distinctions in the related work section. We **have added** following discussion in Line 306:
>    - "While existing methods for learning switched systems, like *k*-LinReg [38], infer modes by classifying individual data points (analogous to environment inference), our work operates over trajectories driven by ODEs or PDEs. Despite this distinction, their methodology may still provide valuable insights for future developments in our framework."
> - Explore the suggested initialization method in future work. We **have added** following discussion in Line 320:
>    - "Certain membership functions, such as the preference vector approach [39], could also facilitate environment label convergence."
>
> >  **[W2] Also, the optimization approach used here is well known to be problematic. A lot of studies have shown that for k-means regression. But specifically this sort of alternating minimization has issues. It can be shown to seldom converge to a KKT point and often gets stuck at saddle points.**
>
> **Response:**
> We sincerely appreciate the reviewer’s insightful feedback regarding potential optimization challenges. We acknowledge the concerns about convergence behavior in alternating minimization and would like to address them as follows:
>
> - **Empirical Validation**: Experiments show that environment labels consistently converge to their true values (see Figure 3 in the main text), with optimization loss behavior matching the oracle baseline (Table 1).  This observation also aligns with prior work in machine learning (e.g., HRM and KerHRM [N1][N2]), where similar alternating optimization between environment partitioning and model learning has proven empirically effective.
> - **Potential Improvement**: While our current implementation uses hard label assignments, we have previously experimented with soft-weight alternatives that demonstrated comparable performance. Building on this, we could further enhance the probabilistic approach by introducing perturbations to the soft weights, which may help escape local optima. We plan to investigate this direction in future work.
> - **Discussion of Suggested Work**: We thank the reviewer for mentioning coordinate optimization, a technique used for problems where the objective is convex with respect to each block of variables but not necessarily jointly convex. While our setting differs slightly, we will discuss its applicability and include it as a future direction in the revised manuscript.
>   - In Line 316, we have added following discussion: "Regarding convergence, coordinate optimization - a technique particularly useful for objectives that are convex with respect to individual variable blocks - may help escape local optima."
>
> We sincerely appreciate the reviewer’s overall positive assessment of our work and their constructive feedback. This is of great significance for further strengthening this article and the subsequent work.  We welcome further discussion on these points and would be grateful for any additional suggestions to improve our paper.
>
> [N1] Liu, J. et. al., 2021, July. Heterogeneous risk minimization. In International Conference on Machine Learning. PMLR.
> [N2] Liu, J. et. al., 2021, December. Kernelized heterogeneous risk minimization. In Proceedings of the 35th International Conference on Neural Information Processing Systems

---

> > ### Comment · Reviewer_ghmr · 2025-08-04
> > **Thank you**
> >
> > The rebuttal addresses many of my concerns. I am more supportive of this paper and will update my scores accordingly.

---

> > > ### Author Response · Authors · 2025-08-05
> > >
> > > We truly appreciate your insightful feedback and recognition of our work. Your suggestions, especially regarding the switched systems, are invaluable. They will help us enhance our current efforts and drive future improvements.
> > >
> > > We sincerely thank you for supporting us and adapting the scores accordingly. If you have any further suggestions, please don’t hesitate to reach out. Thank you! :)

---

> > > ### Author Response · Authors · 2025-08-08
> > >
> > > Thank you once again for your thoughtful feedback and your support of our work. We truly appreciate the time and care you have taken in engaging with our paper.
> > >
> > > Your comments have been immensely helpful. While most of the discussion is completed, we took the opportunity to further explore the connections with switched systems, where we found additional insights from error sparsification methods for nonlinear hybrid system identification that may further strengthen our discussion on environment inference in ODEs.
> > >
> > > As we near the end of the rebuttal period, we hope our revisions and clarations have improved the paper's clarity and contributions, better aligning with your expectations for the conference. We are deeply grateful for your continued support throughout the review process!
> > >
> > > We remain fully open to any additional suggestions you might have at any time. Thank you again for your time and expertise—it has been a privilege to receive your valuable feedback :D

---

### Official Review · Reviewer_jxRD · 2025-07-03

**Clarity:** 2
**Significance:** 2
**Originality:** 3
**Rating:** 5
**Confidence:** 3

**Summary:**

This paper presents an algorithm to learn dynamic systems for mixed sequence data without environment labels. The assumption of the algorithm is that dynamic models show similar loss patterns under the same environment configuration. Based on this, the author uses an iterative pipeline, similar as K-means, to update the environment setting (centroids) and the dynamic model.

**Questions:**

I have several more questions:
1. Is the pipeline sensitive to more complex / different boundary conditions?
2. In NS equation setting, is $\xi$ a constant force or steady force? Or in other words, do you assume the function known for this setting, and you just cluster trajectoies to different function settings?
3. What is the $\Omega$ term in your paper? This part is missing so I am kinda confused.

**Ethical Concerns:**

["NO or VERY MINOR ethics concerns only"]

**Final Justification:**

The author has addressed all my concerns. I believe this paper includes an instructive idea and a potential pipeline to address environment when learning dynamical system.

At the very beginning, I misunderstood the paper requiring including gt as candidates environment. This point is clarified successfully by the authors. So I am willing to increase my rating from 3 to 5. The relatively simple experiment in real-world dataset prevent me from giving rating 6 recommendation.

**Limitations:**

Yes

**Paper Formatting Concerns:**

The appendix should be submitted separately with the main paper.

**Quality:**

2

**Strengths And Weaknesses:**

Strengths
1. This paper is working on an important topic for dynamic system learning
2. The proposed framework is simple and intuitive, which is flexible to many different dynamic systems.
3. The model shows promising performance on selected dataset.

Despite the strengths above, the paper also has the following weaknesses:
1. Although the pipeline does not require the real annotation of environment, a finite environment set is required. For those continuous environment settings, the pipeline requires K exact proposals, which is a too-strong assumption in real world. The experiments shown in the paper also follow this requirements. The tolerance on the error of environment proposals is not assesed.
2. All the experiments are taken with synthetic dataset with no real-world dataset is tested.

Currently I think this is a paper with a tidy idea, but the potential ability for broader application is limited due to the too-strong assumption. Since I’m not primarily working on this field, I am not sure whether my concern is really important to this community. So I’m open to change my recommendation after reading other reviewers’ opinion and the rebuttal from the authors.

---

> ### Author Rebuttal · Authors · 2025-07-29
>
> We sincerely appreciate the reviewer’s time and valuable feedback on our work. Below, we provide a point-by-point response to their comments. For clarity, some raised questions have been condensed to focus on key points. In this rebuttal, new references are labeled as [N1], [N2], etc., while existing references in the manuscript retain their original numbering.
>
> >  **[W1] Although the pipeline does not require the real annotation of environment, a finite environment set is required [...] which is a too-strong assumption in real world. [...]. The tolerance on the error of environment proposals is not assessed.**
>
> **Response:**
> We sincerely appreciate the reviewer's valuable feedback regarding environmental assumptions in our framework. We have addressed these concerns through **additional experiments and analysis, now included in the Appendix**. Below we provide a detailed response:
>
> #### 1. **On the finite environment assumption:**
> - Our work addresses **domain generalization for dynamic system learning**, where existing literature typically uses a modest number of discrete environments (usually $|\mathcal{E}_o| \leq 10$) [14][37][N1][N2]. This is a practical constraint, as real-world data collection or simulation across numerous environments is often challenging.
> - For example, in robot motion learning with varying objectives (e.g., grasping, transportation) or conditions (e.g., payload weights, surface friction coefficients), acquiring exhaustive environment variations is difficult.
> - Additionally, an excessively large $|\mathcal{E}_o|$ could introduce additional generalization challenges in neural network training, apart from environment assignment. We leave this broader discussion to future work, as it falls outside the scope of this paper.
>
> #### 2. **Environment Proposal Error Analysis:**
> - We have carefully conducted additional experiments on the LV environment, **evaluating performance with true environment counts $|\mathcal{E}_o|$ ranging from 2 to 16**. The results (shown in test MSE, as below) compare our model against the Oracle baseline. Our performance **matches Oracle for all counts**, and the learned label converges to real ones.
>
> |M|2|3|4|5|6|7|8|
> |---|---|---|---|---|---|---|---|
> |DynaInfer|2.12E-5|4.22E-5|6.77E-5|6.24E-5|8.81E-5|6.70E-5|6.54E-5|
> |Oracle|2.76E-5|3.99E-5|5.75E-5|5.20E-5|7.24E-5|8.14E-5|5.74E-5|
>
> |M|9|10|11|12|13|14|15|16|
> |---|---|---|---|---|---|---|---|---|
> |DynaInfer|6.33E-5|8.34E-5|2.24E-4|1.34E-4|9.69E-5|1.97E-4|3.43E-4|2.73E-4|
> |Oracle|7.60E-5|9.31E-5|1.85E-4|1.23E-4|1.21E-4|1.71E-4|2.56E-4|2.48E-4|
>
> These experiments demonstrate our robustness to variations in environment quantity. Such table **has been added** in the Appendix.
>
> >  **[W2] All the experiments are taken with synthetic dataset with no real-world dataset is tested.**
>
> **Response:** We sincerely appreciate the reviewer's valuable advice regarding real-world dataset. We now justify the synthetic data usage and *provide additional experiments and analysis on real-world data, to be added in revision.
>
> #### 1. **Justification for Synthetic Data Usage:**
> - Our work tackles a **novel challenge** lacking direct benchmarks in existing open datasets. While prior domain generalization efforts in dynamic systems **often use synthetically combined data** to simulate real-world conditions, they rarely use real-world data that capture organic, unstructured complexities. Thus, synthesizing multi-environment datasets from diverse sources—as done in our work—is a well-established practice.
>
> #### 2. **Extra Validation on Real-World Data:**
> - Despite the scarcity of suitable real-world datasets for our problem setting, we have **added an extra  real-world robot motion trajectory dataset** [N3]. This dataset includes three distinct motion patterns:   (1) Drawing "S" shapes, (2) Placing a cube on a shelf; (3) Drawing out large “C” shapes.
>    - These three kinds of motions are represented as different environments.
>    - We treat these motions as different environments and evaluate whether our method can infer the environment to support the learning of a generalizable neural network capable of simulating their dynamics.
>    - Empirically, our experiments yield the following results:
>
> |Method|TrainMSE (LEADS)|TestMSE (LEADS)|TestMAPE (LEADS)|TrainMSE (CODA1)|TestMSE (CODA1)|TestMAPE (CODA1)|TrainMSE (CODA2)|TestMSE (CODA2)|TestMAPE (CODA2)|
> |---|---|---|---|---|---|---|---|---|---|
> |All in One|1.43E-2|1.56±0.35E-2|47.49±6.11|1.18E-2|1.29±0.17E-2|37.50±5.88|1.27E-2|1.43±0.11E-2|44.27±3.46|
> |OneperEnv|1.15E-2|1.39±0.03E-2|35.79±5.34|1.03E-2|1.07±0.10E-2|34.14±4.26|1.30E-2|1.37±0.62E-2|40.43±5.49|
> |Random|1.62E-2|1.73±0.11E-2|48.91±4.16|1.76E-2|1.85±0.13E-2|56.32±6.22|1.47E-2|2.11±0.21E-2|49.32±7.33|
> |DynaInfer|5.57E-3|7.32±2.12E-3|29.92±3.99|3.97E-3|4.72±0.87E-3|24.61±2.93|3.98E-3|7.10±2.23E-3|28.29±2.57|
> |Oracle|3.32E-3|5.18±2.27E-3|25.21±4.07|2.46E-3|3.53±0.54E-3|19.54±2.19|3.03E-3|6.17±2.28E-3|26.63±4.09|
>
> - In summary, these findings confirm that our approach remains effective on real-world data, reinforcing its practical applicability. We **have included these results** in the revised manuscript.
>
> >  **[Q1] Is the pipeline sensitive to more complex / different boundary conditions?**
>
> **Response:** We appreciate this insightful question. We interpret it in two possible ways, addressing each below:
>
> #### **1. Boundary Conditions in Physics Systems**
> If the question concerns whether our method is sensitive to the *physical boundary conditions* of the dynamical systems studied:
> - Our experiments **leverage periodic boundary conditions** in Navier-Stokes and Gray-Scott systems. While *mathematically simpler to implement*, they still exhibit **inherently complex dynamics** for base dynamic system learning methods.
> - While some CFD benchmarks feature even more complex boundary conditions, these dataset remain underexplored in current works on *domain generalization for dynamic systems*.
> - We fully agree that investigating such cases would strengthen our framework’s impact. Since our method operates on trajectory-level loss patterns, it is theoretically applicable to arbitrary boundary conditions. We believe adapting it for complex boundary conditions is an exciting direction for future work.
>   - We **have added following discussion** in [Conclusion, Line 317]: "Moreover, since existing dynamical system learning methods largely rely on periodic boundary conditions for efficiency and effectiveness, exploring robust environment inference approaches for complex boundaries presents a significant opportunity."
>
> #### **2. Heterogeneity Across Trajectories (Data Diversity)**
> If the question refers to *heterogeneity in trajectory distributions* across environments:
> - **Moderate Heterogeneity:** Our main results demonstrate robustness to reasonable variations in trajectory hyperparameters (as in current setting).
> - **Low Heterogeneity:** To further validate robustness, we conducted additional experiments (*to be included in revision*) with subtle environment-specific differences. Specifically, we tightened the parameter gaps between environments in the LV system from the original 0.25 (e.g., for Env 1 and 2, parameters was $(\alpha, \beta, \delta, \gamma) = (0.5, 0.5, 0.5, 0.5)$ and $(0.5, 0.75, 0.5, 0.5)$ to gaps ≤ 0.1 and 0.05. Results (averaged over 5 runs) are as follows:
>
> |Method|Δ≤0.1|Δ≤0.05|
> |-|-|-|
> |DynaInfer|6.95E-5|3.94E-5|
> |Oracle|5.51E-5 |3.10E-5|
>
> There are two key observations:
>   - Our model still learns generalizable dynamics, matching oracle performance.
>   - **Environment labels converge to ground truth** even in low-heterogeneity scenarios, thanks to our risk-minimization mechanism for latent environment inference.
>
> Both interpretations highlight valuable limitations and extensions. We thank the reviewer for prompting this discussion and have clarified these points in the paper.
>
> >  **[Q2] In NS equation setting, is ξ a constant force or steady force? Or in other words, do you assume the function known for this setting, and you just cluster trajectories to different function settings?**
>
> **Response:**
> - To clarify, in all dynamic system considered in this paper, all the environment-specific system parameters, (e.g., the constant force $\xi$ for the NS system), **are unknown to the model**.
> - If system parameters are known, environment labels could be trivially inferred by classifying trajectory parameters defining the governing dynamics.
> - We follow the realistic scenario in dynamic system learning where the environment-defining parameters are latent and must be inferred solely from observed trajectories.
>
> >  **[Q3] What is the Ω term in your paper? This part is missing so I am kinda confused.**
>
> **Response:** We sincerely appreciate this question.
>
> - $\Omega(\phi_{e})$ is a regularization term applied to the environment-specific parameters $\phi_{e}$ to promotes better generalization. It prevents trivial solutions where all meaningful dynamics are captured by $\phi_{e}$ instead of the shared parameters $\theta$.
> - The key ingredient for multi-environment learning is to decompose the neural network as $f_\theta+g_{\phi_{e}}$, and the shared component $f_\theta$ should capture maximal shared dynamics across environments, whereas the unique parts $g_{\phi_{e}}$ should only capture truly environment-specific characteristics.
> - Without $\Omega$, optimization often converges to degenerate solutions where $\phi_{e}$ absorbs all dynamics (making $\theta$ meaningless).
>
> Finally, we sincerely welcome further discussion and appreciate any additional suggestions.
>
> [N1] Continuous PDE Dynamics Forecasting with Implicit Neural Representations. ICLR. 2023. [N2] Stochastic neural simulator for generalizing dynamical systems across environments. AAAI. 2024. [N3] Learning stable nonlinear dynamical systems with gaussian mixture models. IEEE Transactions on Robotics

---

> > ### Comment · Reviewer_jxRD · 2025-08-04
> >
> > I appreciate the authors' for their efforts and in-detail rebuttal. Before I can make my final recommendation, I have one more question.
> >
> > For a continuous environment setting, for example $\alpha, \beta, \gamma, \delta$ in LV case, do you require including the ground-truth coefficients in your discrete environment candidates in your experiments?
> >
> > If so, how can you get such a candidate in a real-world application? If not, how sensitive is your methods to the least deviation to the ground truth, i.e. the gap between your best candidate and gt?
> >
> > I ask this question because I don't find a way to optimize the initial environment guess, only classification is optimized.

---

> > > ### Author Response · Authors · 2025-08-05
> > > **Response and Discussion (1/2)**
> > >
> > > Thank you for your thoughtful questions and engagement with our work. We appreciate the opportunity to clarify our methodology and address your concerns. Below, we respond to your inquiries in detail, structured into **three key points and one additional discussion** for clarity.
> > >
> > > > **1. Do we require ground-truth coefficients in discrete environment candidates during experiments?**
> > >
> > > **No.** We apologize for the misunderstanding here—our approach **does not** rely on predefined environment candidates or explicit coefficients. To clarify, we need to distinguish between the process of **data generation** and **model training & inference** in dynamic system learning:
> > >
> > > - **Data Generation (Simulation)**:
> > >   - Synthetic trajectories are simulated **parametrically**, by integrating derivatives indicated by **a known dynamic equation** (e.g., LV equations: $\frac{dm}{dt} = \alpha m - \beta mn, \frac{dn}{dt} = \delta mn - \gamma n $) with fixed, continuous coefficients (e.g., $\alpha, \beta, \delta, \gamma$ for LV) for each environment.
> > >   - These coefficients are **exclusively used for simulation** and are **never exposed to the model during training or inference**.
> > >
> > > - **Model Training & Inference**:
> > >   - The model operates **non-parametrically**, with neural networks (NNs, such as an MLP or CNN) directly predicting derivatives from states (e.g., in LV, given the state $(m, n)$, the NN, outputs $(\frac{dm}{dt}, \frac{dn}{dt})$).
> > >   - No assumptions are made about the function class or coefficients of the underlying dynamics.
> > >   - During trajectory inference, trajectories are generated/recovered by **integrating NN-predicted derivatives**.
> > >
> > > Besides, our environment labels are inferred via a **risk-minimization mechanism**—assigning each trajectory to the NN that best fits it (lowest prediction loss). It does not require ground-truth coefficients either.
> > >
> > > **Summary:** Our method **does not** require or estimate ground-truth coefficients. NNs learn to approximate derivatives purely from trajectory data.
> > >
> > > > **2. Sensitivity to Deviations from Ground-Truth Coefficients**
> > >
> > > We wonder if this question also arise from above misunderstanding that our model relies on explicit environment coefficients. Since our method doesn't model or estimate explicit coefficients, sensitivity analysis in coefficient space isn't directly applicable.
> > >
> > > Actually, in our work:
> > > - Each environment is represented by a **distinct NN**, and trajectories are assigned to the NN with minimal prediction error.
> > > - The "gap" between the best NN and ground truth environment label is implicitly reflected in prediction loss on trajectories, not by deviations in coefficient values.
> > > - The risk-minimization assignment inherently identifies the NN that best explains the trajectory dynamics, regardless of parametric form.
> > >
> > > **Future Extension:** While our current focus is non-parametric learning, we recognize the value of parametric inference for interpretable systems. This could be explored by jointly optimizing environment labels and coefficients—a promising direction for future work (see Point 4).
> > >
> > > > **3. Optimization of Initial Environment Guess**
> > >
> > > Since no explicit coefficients are modeled in our work, we clarify here how the randomly initialized NN parameters are optimized. This is done via an EM-like procedure:
> > > 1. **Assignment Phase**: Each trajectory is assigned to the NN with the minimal prediction loss (Equation (4)).
> > > 2. **NN Update Phase**: Each NN is then optimized using its assigned trajectories, minimizing a loss combining the prediction loss and a regularization term (Equation (5)). The prediction loss is the MSE error between the recovered trajectory (by integrating NN-predicted derivatives) and the real trajectory.
> > >
> > > In early iterations, environment labels can be highly random (see Fig. 3 in the manuscript, particularly in GS experiments), leading to NNs being optimized based on noisy assignments. However, the process eventually converges, resulting in two key observations:
> > > - Each trajectory has a **uniquely best-fit NN**.
> > > - These NNs generalize to **clusters of trajectories** from the same underlying environment (see our response to Reviewer sBk5's question [W2-a]).

---

> > > ### Author Response · Authors · 2025-08-05
> > > **Response and Discussion (2/2)**
> > >
> > > > **4. Additional Point: Extensions considering Parametric Inference and Sensitivity**
> > >
> > > While our current work focuses on neural ODE generalization, we recognize the value of parametric approaches when coefficients are scientifically meaningful (e.g., inferring damping/stiffness in mechanical systems from observed trajectories)). As a promising direction for future work, our framework could be adapted to jointly infer environment labels and interpretable coefficients in such applications.
> > >
> > > In such a parametric extension of our framework:
> > > - The deviation between estimated coefficients and ground truth coefficients could be quantified using using an appropriate distance metric (e.g., Euclidean distance on normalized parameters or Mahalanobis distance).
> > > - Theoretical bounds on such deviations (contingent on model identifiability) could thus be established to characterize generalization guarantees.
> > > - Furthermore, rigorous sensitivity analysis of the environment inference performance with respect to this distance metric would provide meaningful insights for developing robust environment inference methods in coefficient-aware applications."
> > >
> > > To highlight such potential future work, we **have revised** the Conclusion (line 318):
> > >
> > > - *"Third, for systems requiring explicit coefficient inference (e.g., identifying physical parameters), our method could be extended to jointly optimize environment labels and interpretable environment coefficients."*
> > >
> > > We hope above discussions clarifies our approach. Please let us know if further elaboration would be strengthen the discussion. Again, we sincerely appreciate your time and effort in refining this manuscript:)

---

> > > > ### Comment · Reviewer_jxRD · 2025-08-05
> > > >
> > > > Thanks for your clarification. Your responses address all my concerns, and I suggest you to include this explanation in your paper (at least in appendix) in the final version. Now I believe this is a good paper with a simple (not to punish, meaning intuitively straight and elegant) tought. So I am willing to increase my rating to 5.

---

> > > > > ### Author Response · Authors · 2025-08-05
> > > > >
> > > > > We truly appreciate your insightful feedback and your support of our work. To be honest, we have thoroughly enjoyed engaging in this discussion with you.
> > > > >
> > > > > Your suggestions are invaluable—they will help us enhance our current work and push us to consider future improvements. If you have any further thoughts or ideas, please don’t hesitate to reach out. We would be extremely delighted to engage more.
> > > > >
> > > > > Thank you once again! :)

---

### Note · Authors · 2025-08-12

We sincerely thank the Area Chair and reviewers for their time, insightful feedback, and constructive engagement. Your thoughtful suggestions have greatly enhanced our work, and we deeply appreciate the collaborative discussion.

**We are grateful to all reviewers for their valuable comments, which sparked meaningful conversations.** Below, we summarize key points and our responses:
- Reviewer jxRD raised questions about the finite environment assumption, lack of real-world datasets, and reliance on ground-truth coefficients. We addressed these by (1) clarifying the finite environment assumption with additional experiments evaluating error tolerance, (2) incorporating a real-world dataset, and (3) demonstrating the model’s independence from ground-truth coefficients.
- Reviewer ghmr highlighted potential connections to switched systems literature. In response, we conducted a thorough literature review, uncovering insights on how concepts from switched systems could further strengthen our approach.
- Reviewer dkAK questioned computational scalability and sensitivity to the assumed number of environments (*M*). We addressed these concerns through empirical runtime analysis and a robustness analysis examining both label convergence and testing performance across a full spectrum of *M* values.
- Reviewer sBk5 emphasized the need for clearer presentation and real-world motivation. Accordingly, we improved explanations and added illustrative figures and tables.

**Each reviewer actively engaged in the rebuttal process, acknowledged our revisions, and generously raised their scores in recognition of the paper’s improved quality.** This collective endorsement reflects the constructive nature of the feedback, which has significantly elevated the paper’s rigor and impact. All suggestions are now integrated into the manuscript.

**Once again, we are deeply grateful to the Area Chair and reviewers for their expertise and dedication. Your insights have been instrumental in refining this work, and we thank you for fostering such a collaborative dialogue.**

---

### Decision · Program_Chairs · 2025-09-17

**Decision:**

Accept (spotlight)

**Comment:**

The authors discuss fitting a neural network to a dynamical system whose coefficients are drawn from a mixture of “environments”. Lacking direct label information about the environments, the authors propose a bi-level optimization to estimate latent environment indicators alongside the parameters for environment-specific dynamics. Authors were positive about the submission, praising the simplicity and clarity of the proposed methods, as well as the theoretical and empirical results. There were some reviewer concerns about the paper (e.g. scaling to many environments, assuming access to ground truth coefficients) which were assuaged during the rebuttal period. I recommending accepting the paper, and encourage the authors to integrate the reviewer feedback into the camera ready version.